# Convex-Concave Min-Max Stackelberg Games

**Denizalp Goktas**
Department of Computer Science
Brown University
Providence, RI 02912
denizalp_goktas@brown.edu

**Amy Greenwald**
Department of Computer Science
Brown University
Providence, RI 02912
amy_greenwald@brown.edu

## Abstract

Min-max optimization problems (i.e., min-max games) have been attracting a great deal of attention because of their applicability to a wide range of machine learning problems. Although significant progress has been made recently, the literature to date has focused on games with independent strategy sets; little is known about solving games with dependent strategy sets, which can be characterized as min-max Stackelberg games. We introduce two first-order methods that solve a large class of convex-concave min-max Stackelberg games, and show that our methods converge in polynomial time. Min-max Stackelberg games were first studied by Wald, under the posthumous name of Wald's maximin model, a variant of which is the main paradigm used in robust optimization, which means that our methods can likewise solve many convex robust optimization problems. We observe that the computation of competitive equilibria in Fisher markets also comprises a min-max Stackelberg game. Further, we demonstrate the efficacy and efficiency of our algorithms in practice by computing competitive equilibria in Fisher markets with varying utility structures. Our experiments suggest potential ways to extend our theoretical results, by demonstrating how different smoothness properties can affect the convergence rate of our algorithms.

## 1 Introduction

Min-max optimization problems have attracted a great deal of attention recently because of their applicability to a wide range of machine learning problems. Examples of settings in which min-max optimization problems arise include, but are not limited to, reinforcement learning [15], generative adversarial networks [66], fairness in machine learning [14, 23, 48, 68, 82], adversarial learning [72], generative adversarial imitation learning [11, 32], and statistical learning (e.g., learning parameters of exponential families) [14]. These applications often require solving a **constrained min-max optimization problem** (with independent feasible sets): i.e., $\min_{\boldsymbol{x} \in X} \max_{\boldsymbol{y} \in Y} f(\boldsymbol{x}, \boldsymbol{y})$, where $f : X \times Y \to \mathbb{R}$ is continuous, and $X \subset \mathbb{R}^n$ and $Y \subset \mathbb{R}^m$ are non-empty and compact.

A **convex-concave** constrained min-max optimization problem is one in which $f$ is convex in $\boldsymbol{x}$ and concave in $\boldsymbol{y}$. In the special case of convex-concave objective functions, the seminal minimax theorem holds: i.e., $\min_{\boldsymbol{x} \in X} \max_{\boldsymbol{y} \in Y} f(\boldsymbol{x}, \boldsymbol{y}) = \max_{\boldsymbol{y} \in Y} \min_{\boldsymbol{x} \in X} f(\boldsymbol{x}, \boldsymbol{y})$ [55]. This theorem guarantees the existence of a **saddle point** (i.e., a point that is simultaneously a minimum of $f$ in the $\boldsymbol{x}$-direction and a maximum of $f$ in the $\boldsymbol{y}$-direction), and allows us to interpret the optimization problem as a simultaneous-move, zero-sum game, where $\boldsymbol{y}^*$ (resp. $\boldsymbol{x}^*$) is a best-response of the outer (resp. inner) player to their opponent's strategy $\boldsymbol{x}^*$ (resp. $\boldsymbol{y}^*$), in which case a saddle point is also called a minimax point or a **Nash equilibrium**.

In this paper, we show that the computation of competitive equilibria in Fisher markets [10], a central solution concept for one of the most well-studied market models in algorithmic game theory [57], can be understood as a convex-concave min-max optimization problem, albeit with *dependent*

feasible sets. We define a **constrained min-max optimization problem with dependent feasible sets** as an optimization problem of the following form: $\min_{\boldsymbol{x} \in X} \max_{\boldsymbol{y} \in Y : \boldsymbol{g}(\boldsymbol{x}, \boldsymbol{y}) \geq \boldsymbol{0}} f(\boldsymbol{x}, \boldsymbol{y})$, where $f : X \times Y \to \mathbb{R}$ is continuous, $X \subset \mathbb{R}^n$ and $Y \subset \mathbb{R}^m$ are non-empty and compact, and $\boldsymbol{g}(\boldsymbol{x}, \boldsymbol{y}) = (g_1(\boldsymbol{x}, \boldsymbol{y}), \ldots, g_K(\boldsymbol{x}, \boldsymbol{y}))^T$ with $g_k : X \times Y \to \mathbb{R}$.[1] Unfortunately, certain desirable properties of standard convex-concave min-max optimization problems with independent feasible sets do not carry over to the more general dependent setting. First and foremost, the minimax theorem [55] does not hold, which in turn precludes the existence of Nash equilibrium in general:

**Example 1.1.** *Consider the following constrained min-max optimization problem with dependent feasible sets:* $\min_{x \in [-1,1]} \max_{y \in [-1,1] : x+y \leq 0} x^2 + y + 1$. *The optimum is* $x^* = 1/2, y^* = -1/2$, *with value* $3/4$. *Now, consider the same problem, with the order of the* min *and the* max *reversed:* $\max_{y \in [-1,1]} \min_{x \in [-1,1] : x+y \leq 0} x^2 + y + 1$. *The optimum is now* $x^* = -1, y^* = 1$, *with value* 3.

Without a minimax theorem, constrained min-max optimization problems with dependent feasible sets are more appropriately viewed as two-player sequential, i.e., Stackelberg, games where the outer player chooses $\boldsymbol{x} \in X$ before the inner player responds with their choice of $\boldsymbol{y}(\boldsymbol{x}) \in Y$ s.t. $\boldsymbol{g}(\boldsymbol{x}, \boldsymbol{y}(\boldsymbol{x})) \geq \boldsymbol{0}$. The relevant equilibrium concept is then a Stackelberg [78] equilibrium,[2] in which the outer player optimizes their choice assuming the inner player will best-respond: i.e., optimize their choice in turn. We thus refer to constrained min-max optimization problems with dependent feasible sets as **min-max Stackelberg games**. In such games, the **value function**[3] $V : X \to \mathbb{R}$ is defined as $V(\boldsymbol{x}) = \max_{\boldsymbol{y} \in Y : \boldsymbol{g}(\boldsymbol{x}, \boldsymbol{y}) \geq \boldsymbol{0}} f(\boldsymbol{x}, \boldsymbol{y})$. This function represents the outer player's loss, assuming the inner player chooses a feasible best-response, so it is the function the outer player seeks to minimize. The inner player seeks only to maximize the objective function. The first-order necessary and sufficient conditions for a tuple $(\boldsymbol{x}^*, \boldsymbol{y}^*) \in X \times Y$ to be a Stackelberg equilibrium in a convex-concave min-max Stackelberg game are given by the KKT stationarity conditions for the two players' optimization problems, namely $\min_{\boldsymbol{x} \in X} V(\boldsymbol{x})$ for the outer player, and $\max_{\boldsymbol{y} \in Y : \boldsymbol{g}(\boldsymbol{x}, \boldsymbol{y}) \geq \boldsymbol{0}} f(\boldsymbol{x}, \boldsymbol{y})$ for the inner player.

In the independent strategy setting, Danskin's theorem [17] states that $\nabla_{\boldsymbol{x}} V(\boldsymbol{x}) = \nabla_{\boldsymbol{x}} f(\boldsymbol{x}, \boldsymbol{y}^*)$, where $\boldsymbol{y}^* \in \arg\max_{\boldsymbol{y} \in Y} f(\boldsymbol{x}, \boldsymbol{y})$. In other words, when there is no dependence among the players' strategy sets, the gradient of the value function coincides with that of the objective function. The first-order necessary and sufficient conditions for a tuple $(\boldsymbol{x}^*, \boldsymbol{y}^*)$ to be an (interior) saddle point are for it to be a stationary point of $f$ (i.e., $\nabla_{\boldsymbol{x}} f(\boldsymbol{x}^*, \boldsymbol{y}^*) = \nabla_{\boldsymbol{y}} f(\boldsymbol{x}^*, \boldsymbol{y}^*) = 0$). It is therefore logical for players in the independent/simultaneous-move setting to follow the gradient of the objective function. In the dependent/sequential setting, however, the direction of steepest descent (resp. ascent) for the outer (resp. inner) player is the gradient of their *value* function.

**Example 1.2.** *Consider, once again, the problem posed in Example 1.1, and recall Jin, Netrapalli, and Jordan's gradient descent with max-oracle algorithm [41]:* $\boldsymbol{x}^{(t+1)} = \boldsymbol{x}^{(t)} - \eta \nabla_{\boldsymbol{x}} f\left(\boldsymbol{x}^t, \boldsymbol{y}^*(\boldsymbol{x}^{(t)})\right)$, *where* $\boldsymbol{y}^*(\boldsymbol{x}^{(t)}) \in \arg\max_{\boldsymbol{y} \in Y : \boldsymbol{g}(\boldsymbol{x}^{(t)}, \boldsymbol{y}) \geq 0} f(\boldsymbol{x}^{(t)}, \boldsymbol{y})$ *for* $\eta > 0$. *Applied to this sample problem, with* $\eta = 1$, *this algorithm yields the following update rule:* $x^{(t+1)} = x^{(t)} - 2x^{(t)} = -x^{(t)}$. *Thus, letting* $x^{(0)}$ *equal any feasible* $x$, *the output cycles between* $x$ *and* $-x$, *so that the average of the iterates converges to* $x^* = 0$ *(with* $y^* = 0$*), which is not a Stackelberg equilibrium, as the Stackelberg equilibrium of this game is* $x^* = 1/2, y^* = -1/2$.

*Now consider an algorithm that updates based not on gradient of the objective function, but of the value function, namely* $\boldsymbol{x}^{(t+1)} = \boldsymbol{x}^{(t)} - \eta \nabla_{\boldsymbol{x}} V(\boldsymbol{x}^t)$. *The value function is* $V(x) = \max_{y \in [-1,1] : x+y \leq 0} x^2 + y + 1 = x^2 - x + 1$, *with gradient* $V'(x) = 2x - 1$. *Thus, when* $\eta = 1$, *this algorithm yields the following update rule:* $x^{(t+1)} = x^{(t)} - V'(x^{(t)}) = x^{(t)} - 2x^{(t)} + 1 = -x^{(t)} + 1$. *If we run this algorithm from initial point* $x^{(0)} = 1/8$, *we get* $x^{(1)} = -1/8 + 1 = 7/8$, $x^{(2)} = -7/8 + 1 = 1/8$, *and so on. The average of the iterates* $1/8, 7/8, 1/8, \ldots$ *converges to* $x^* = 1/2 (1/8 + 7/8) = 1/2$, *and correspondingly* $y^* = -1/2$, *which is indeed the Stackelberg equilibrium.*

---

[1]Although notationally similar, we note that this model is more general than that of Daskalakis, Skoulakis, and Zampetakis [18]. The authors state that under the assumptions of the minimax theorem [55], a Nash equilibrium is guaranteed to exist in their model. When there is dependence among the players' feasible sets, however, this claim does not hold (see Example 1.1).

[2]In constrained convex-concave min-max optimization problems with independent feasible sets, the notions of Nash and Stackelberg equilibria coincide. They also coincide in the dependent setting, *when Nash equilibria exist*, but they do not coincide in general, since Nash equilibria need not exist.

[3]Note that this use of the term value function in economics is distinct from its use in reinforcement learning.

*In this paper*, we introduce two first-order (subgradient) methods that solve min-max Stackelberg games—to our knowledge the first such methods. Our approach relies on a new generalization of a series of fundamental results in mathematical economics known as **envelope theorems** [1, 50]. Envelope theorems generalize aspects of Danskin's theorem, by providing explicit formulas for the gradient of the value function in dependent strategy settings, *when a derivative is guaranteed to exist*. To sidestep the differentiability issue, we introduce a generalized envelope theorem that gives an explicit formula for the subdifferential of the value function in dependent strategy settings.

Our first algorithm follows Jin, Netrapalli, and Jordan [41], assuming access to a max-oracle that returns $\boldsymbol{y}^* \in \arg\max_{\boldsymbol{y} \in Y : \boldsymbol{g}(\boldsymbol{x}, \boldsymbol{y}) \geq 0} f(\boldsymbol{x}, \boldsymbol{y})$, given $\boldsymbol{x} \in X$. Hence, our first algorithm solves only for an optimal $\boldsymbol{x}^*$, while our second algorithm explicitly solves for both $\boldsymbol{x}^*$ and $\boldsymbol{y}^*$. We show that both algorithms converge in polynomial time to a Stackelberg equilibrium of any convex-concave min-max Stackelberg game with nonempty, compact and convex dependent strategy sets. In Table 1, we summarize the iteration complexities of our algorithms, i.e., the number of iterations required to achieve an $\varepsilon$-approximate equilibrium, where $\varepsilon$ is the desired precision.

Table 1: Iteration complexities of Algorithms 1 and 2 for min-max Stackelberg games. Here, $\mu_{\boldsymbol{x}}$ and $\mu_{\boldsymbol{y}}$ are strong convexity/concavity parameters.

| Properties of $f$ | Iteration Complexity | |
| --- | --- | --- |
| | Algorithm 1 | Algorithm 2 |
| $\mu_{\boldsymbol{x}}$-Strongly-Convex-$\mu_{\boldsymbol{y}}$-Strongly-Concave | $O\left(\varepsilon^{-1}\right)$ | $\tilde{O}(\varepsilon^{-1})$ |
| $\mu_{\boldsymbol{x}}$-Strongly-Convex-Concave | | $O(\varepsilon^{-2})$ |
| Convex-$\mu_{\boldsymbol{y}}$-Strongly-Concave | $O\left(\varepsilon^{-2}\right)$ | $\tilde{O}(\varepsilon^{-2})$ |
| Convex-Concave | | $O(\varepsilon^{-3})$ |

Finally, we apply our results to the computation of competitive equilibria in Fisher markets. In this context, our method for solving a min-max Stackelberg game reduces to solving the market in a decentralized manner using a natural market dynamic called tâtonnement [80]. We demonstrate the efficacy and efficiency of our algorithms in practice by running a series of experiments in which we compute competitive equilibria in Fisher markets with varying utility structures—specifically, linear, Cobb-Douglas, and Leontief. Although our theoretical results do not apply to all these Fisher markets—Leontief utilities, in particular, are not differentiable—tâtonnement converges in all our experiments. That said, the rate of convergence does seem to depend on the smoothness characteristics of the utility structures; we observe slower convergence for Leontief utilities, and faster convergence than our theory predicts for Cobb-Douglas utilities, which are not only differentiable, but whose value function is also differentiable.

**Related Work** Our model of min-max Stackelberg games seems to have first been studied by Wald, under the posthumous name of Wald's maximin model [79]. A variant of Wald's maximin model is the main paradigm used in robust optimization, a fundamental framework in operations research for which many methods have been proposed [6, 56, 61]. Shimizu and Aiyoshi [70, 71] proposed the first algorithm to solve min-max Stackelberg games via a relaxation to a constrained optimization problem with infinitely many constraints, which nonetheless seems to perform well in practice. More recently, Segundo, Krohling, and Cosme [69] proposed an evolutionary algorithm for these games, but they provided no guarantees. As pointed out by Postek and Shtern, all prior methods either require oracles and are stochastic in nature [6], or rely on a binary search for the optimal value, which can be computationally complex [56]. The algorithms we propose in this paper circumvent the aforementioned issues and can be used to solve a large class of convex robust optimization problems in a simple and efficient manner.

Extensive-form games in which players' strategy sets can depend on other players' actions have been studied by Davis, Waugh, and Bowling [19] assuming payoffs are bilinear, and by Farina, Kroer, and Sandholm [28] for another specific class of convex-concave payoffs. Fabiani et al. [25] and Kebriaei and Iannelli [43] study more general settings than ours, namely non-zero-sum Stackelberg games with more than two players. Both sets of authors derive convergence guarantees assuming specific payoff structures, but their algorithms do not converge in polynomial time.

Min-max Stackelberg games naturally model various economic settings. They are related to the abstract economies first studied by Arrow and Debreu [4]; however, the solution concept that has been the focus of this literature is generalized Nash equilibrium [26, 27], which, like Stackelberg, is a weaker solution concept than Nash, but which makes the arguably unreasonable assumption that the players move simultaneously and nonetheless satisfy the constraint dependencies on their strategies imposed by one another's moves. See Appendix A for a more detailed discussion of generalized Nash equilibria versus Stackelberg equilibria.

Duetting et al. [22] study optimal auction design problems. They propose a neural network architecture called RegretNet that represents optimal auctions, and train their networks using Algorithm 3.4. Optimal auction design problems can be seen as min-max Stackelberg games; however, as their objectives are non-convex-concave in general, our guarantees do not apply.

In this paper, we observe that solving for the competitive equilibrium of a Fisher market can also be seen as solving a (convex-concave) min-max Stackelberg game. The study of the computation of competitive equilibria in Fisher markets was initiated by Devanur et al. [20], who provided a polynomial-time method for the case of linear utilities. Jain, Vazirani, and Ye [40] subsequently showed that a large class of Fisher markets could be solved in polynomial-time using interior point methods. Recently, Gao and Kroer [29] studied an alternative family of first-order methods for solving Fisher markets (only; not min-max Stackelberg games more generally), assuming linear, quasilinear, and Leontief utilities, as such methods can be more efficient when markets are large.

See Appendix G for a detailed discussion of recent progress on solving min-max Stackelberg games, both in the convex-concave case and the non-convex-concave case.

## 2 Preliminaries

**Notation** We use Roman uppercase letters to denote sets (e.g., $X$), bold uppercase letters to denote matrices (e.g., $\boldsymbol{X}$), bold lowercase letters to denote vectors (e.g., $\boldsymbol{p}$), and Roman lowercase letters to denote scalar quantities, (e.g., $c$). We denote the $i$th row vector of a matrix (e.g., $\boldsymbol{X}$) by the corresponding bold lowercase letter with subscript $i$ (e.g., $\boldsymbol{x}_i$). Similarly, we denote the $j$th entry of a vector (e.g., $\boldsymbol{p}$ or $\boldsymbol{x}_i$) by the corresponding Roman lowercase letter with subscript $j$ (e.g., $p_j$ or $x_{ij}$). We denote the set of integers $\{1, \ldots, n\}$ by $[n]$, the set of natural numbers by $\mathbb{N}$, the set of real numbers by $\mathbb{R}$, the set of non-negative real numbers by $\mathbb{R}_+$. We denote by $\Pi_Y$ the Euclidean projection operator onto the set $Y \subset \mathbb{R}^m$: i.e., $\Pi_Y(\boldsymbol{y}) = \arg\min_{\boldsymbol{z} \in Y} \|\boldsymbol{y} - \boldsymbol{z}\|_2$.

**A min-max Stackelberg game**, denoted $(X, Y, f, \boldsymbol{g})$, is a two-player, zero-sum game, where one player, who we call the $\boldsymbol{x}$-player (resp. the $\boldsymbol{y}$-player), is trying to minimize their loss (resp. maximize their gain), defined by a continuous **objective function** $f : X \times Y \to \mathbb{R}$, by choosing a strategy from a compact **strategy set** $X \subset \mathbb{R}^n$ (resp. $Y \subset \mathbb{R}^m$), and s.t. $\boldsymbol{g}(\boldsymbol{x}, \boldsymbol{y}) \geq 0$ where $\boldsymbol{g}(\boldsymbol{x}, \boldsymbol{y}) = (g_1(\boldsymbol{x}, \boldsymbol{y}), \ldots, g_K(\boldsymbol{x}, \boldsymbol{y}))^T$ with $g_k : X \times Y \to \mathbb{R}$. A strategy profile $(\boldsymbol{x}, \boldsymbol{y}) \in X \times Y$ is said to be **feasible** iff for all $k \in [K]$, $g_k(\boldsymbol{x}, \boldsymbol{y}) \geq 0$. The function $f$ maps a pair of feasible strategies taken by the players $(\boldsymbol{x}, \boldsymbol{y}) \in X \times Y$ to a real value (i.e., a payoff), which represents the loss (resp. the gain) of the $\boldsymbol{x}$-player (resp. $\boldsymbol{y}$-player). A min-max game is said to be convex-concave if the objective function $f$ is convex-concave.

The relevant solution concept for Stackelberg games is the **Stackelberg equilibrium**:

**Definition 2.1** (Stackelberg Equilibrium). *Consider the min-max Stackelberg game $(X, Y, f, \boldsymbol{g})$. A strategy profile $(\boldsymbol{x}^*, \boldsymbol{y}^*) \in X \times Y$ such that $\boldsymbol{g}(\boldsymbol{x}^*, \boldsymbol{y}^*) \geq \boldsymbol{0}$ is a $(\varepsilon, \delta)$-Stackelberg equilibrium if $\max_{\boldsymbol{y} \in Y : \boldsymbol{g}(\boldsymbol{x}^*, \boldsymbol{y}) \geq 0} f(\boldsymbol{x}^*, \boldsymbol{y}) - \delta \leq f(\boldsymbol{x}^*, \boldsymbol{y}^*) \leq \min_{\boldsymbol{x} \in X} \max_{\boldsymbol{y} \in Y : \boldsymbol{g}(\boldsymbol{x}, \boldsymbol{y}) \geq 0} f(\boldsymbol{x}, \boldsymbol{y}) + \varepsilon.$*

Intuitively, a $(\varepsilon, \delta)$-Stackelberg equilibrium is a point at which the $\boldsymbol{x}$-player's (resp. $\boldsymbol{y}$-player's) payoff is no more than $\varepsilon$ (resp. $\delta$) away from its optimum. By Proposition B.2, a $(0, 0)$-Stackelberg equilibrium is guaranteed to exist in min-max Stackelberg, its value is unique, and by Proposition B.3 the set of Stackelberg equilibria is compact and convex.

Finally, we define several concepts that are needed in our convergence proofs. Given $A \subset \mathbb{R}^n$, the function $f : A \to \mathbb{R}$ is said to be $\ell_f$-**Lipschitz-continuous** iff $\forall \boldsymbol{x}_1, \boldsymbol{x}_2 \in X, \|f(\boldsymbol{x}_1) - f(\boldsymbol{x}_2)\| \leq \ell_f \|\boldsymbol{x}_1 - \boldsymbol{x}_2\|$. If the gradient of $f$, $\nabla f$, is $\ell_{\nabla f}$-Lipschitz-continuous, we then refer to $f$ as $\ell_{\nabla f}$-**Lipschitz-smooth**. A function $f : A \to \mathbb{R}$ is $\mu$-**strongly convex** if $f(\boldsymbol{x}_1) \geq f(\boldsymbol{x}_2) + \langle \nabla_{\boldsymbol{x}} f(\boldsymbol{x}_2), \boldsymbol{x}_1 - \boldsymbol{x}_2 \rangle + \mu/2 \|\boldsymbol{x}_1 - \boldsymbol{x}_1\|^2$, and $\mu$-**strongly concave** if $-f$ is $\mu$-strongly convex.

# 3   First-Order Methods via an Envelope Theorem

The envelope theorems, popular tools in mathematical economics, allow for explicit formulas for the gradient of the value function in min-max games, even when the strategy sets are dependent. Afriat [1] appears to have been the first make use of the Lagrangian to differentiate the value function, though his conclusion was later obtained under weaker assumptions by Milgrom and Segal [50]. Our envelope theorem relies on the following assumptions:

**Assumption 3.1.** *1. $f, g_1, \ldots, g_K$ are continuous and convex-concave; 2. $\nabla_{\boldsymbol{x}} f, \nabla_{\boldsymbol{x}} g_1, \ldots, \nabla_{\boldsymbol{x}} g_K$ are continuous; and 3. (Slater's condition) $\forall \boldsymbol{x} \in X, \exists \widehat{\boldsymbol{y}} \in Y$ s.t. $g_k(\boldsymbol{x}, \widehat{\boldsymbol{y}}) > 0$, for all $k = 1, \ldots, K$.*

Milgrom and Segal's [50] envelope theorem provides an explicit formula for the gradient of the value function. When strategy sets are independent, under mild assumptions, this function is guaranteed to be differentiable [58]. When strategy sets are dependent, however, it is not necessarily differentiable, as seen in Example C.2. As a remedy, we present a subdifferential envelope theorem for non-differentiable value functions:

**Theorem 3.2** (Subdifferential Envelope Theorem). *Consider the value function $V(\boldsymbol{x}) = \max_{\boldsymbol{y} \in Y : \boldsymbol{g}(\boldsymbol{x}, \boldsymbol{y}) \geq 0} f(\boldsymbol{x}, \boldsymbol{y})$. Let $Y^*(\boldsymbol{x}) = \arg\max_{\boldsymbol{y} \in Y : \boldsymbol{g}(\boldsymbol{x}, \boldsymbol{y}) \geq \mathbf{0}} f(\boldsymbol{x}, \boldsymbol{y})$ and suppose Assumption 3.1 holds. Then, at any point $\widehat{\boldsymbol{x}} \in X$, $\partial_{\boldsymbol{x}} V(\widehat{\boldsymbol{x}}) =*

$$
\mathrm{conv} \left( \bigcup_{\boldsymbol{y}^*(\widehat{\boldsymbol{x}}) \in Y^*(\widehat{\boldsymbol{x}})} \bigcup_{\lambda_k^*(\widehat{\boldsymbol{x}}, \boldsymbol{y}^*(\widehat{\boldsymbol{x}})) \in \Lambda^*(\widehat{\boldsymbol{x}}, \boldsymbol{y}^*(\widehat{\boldsymbol{x}}))} \left\{ \nabla_{\boldsymbol{x}} f\left(\widehat{\boldsymbol{x}}, \boldsymbol{y}^*(\widehat{\boldsymbol{x}})\right) + \sum_{k=1}^{K} \lambda_k^*(\widehat{\boldsymbol{x}}, \boldsymbol{y}^*(\widehat{\boldsymbol{x}})) \nabla_{\boldsymbol{x}} g_k\left(\widehat{\boldsymbol{x}}, \boldsymbol{y}^*(\widehat{\boldsymbol{x}})\right) \right\} \right) ,
$$

(1)

*where $\mathrm{conv}$ is the convex hull operator and $\boldsymbol{\lambda}^*(\widehat{\boldsymbol{x}}, \boldsymbol{y}^*(\widehat{\boldsymbol{x}})) = (\lambda_1^*(\widehat{\boldsymbol{x}}, \boldsymbol{y}^*(\widehat{\boldsymbol{x}})), \ldots, \lambda_K^*(\widehat{\boldsymbol{x}}, \boldsymbol{y}^*(\widehat{\boldsymbol{x}})))^T \in \Lambda^*(\widehat{\boldsymbol{x}}, \boldsymbol{y}^*(\widehat{\boldsymbol{x}}))$ are the optimal KKT multipliers associated with $\boldsymbol{y}^*(\widehat{\boldsymbol{x}}) \in Y^*(\widehat{\boldsymbol{x}})$.*

The envelope theorem states that the gradient of a differentiable value function is the gradient of the Lagrangian evaluated at the optimal solution. Generalizing this fact, our subdifferential envelope theorem states that every subgradient of the value function, $V(\boldsymbol{x}) = \max_{\boldsymbol{y} \in Y : \boldsymbol{g}(\boldsymbol{x}, \boldsymbol{y}) \geq 0} f(\boldsymbol{x}, \boldsymbol{y})$ is a convex combination of the values of the gradient of the Lagrangian evaluated at the optimal solutions $(\boldsymbol{y}^*(\boldsymbol{x}), \boldsymbol{\lambda}^*(\boldsymbol{x}, \boldsymbol{y}^*(\boldsymbol{x}))) \in Y^*(\boldsymbol{x}) \times \Lambda^*(\boldsymbol{x}, \boldsymbol{y}^*(\boldsymbol{x}))$.

With our envelope theorem in hand, we are now ready to present two gradient-descent/ascent-type algorithms for min-max Stackelberg games, which follow the gradient of the value function.

Our first algorithm, **max-oracle gradient-descent**, following Jin, Netrapalli, and Jordan [41], assumes access to a max-oracle, which given $\boldsymbol{x} \in X$, returns a $\delta$-maximum of the inner player's value function. That is, for all $\boldsymbol{x} \in X$, the max-oracle returns $\widehat{\boldsymbol{y}} \in Y$ s.t. $\boldsymbol{g}(\boldsymbol{x}, \widehat{\boldsymbol{y}}) \geq 0$ and $f(\boldsymbol{x}, \widehat{\boldsymbol{y}}) \geq \max_{\boldsymbol{y} \in Y : \boldsymbol{g}(\boldsymbol{x}, \boldsymbol{y}) \geq 0} f(\boldsymbol{x}, \boldsymbol{y}) - \delta$. It then runs (sub)gradient descent on the outer player's value function, using Theorem 3.2 to compute the requisite subgradients. Inspired by the multi-step gradient-descent algorithm of Nouiehed et al. [58], our second algorithm, **nested gradient-descent/ascent** (see Appendix E), computes both $\boldsymbol{x}^*$ and $\boldsymbol{y}^*$ explicitly, without oracle access. We simply replace the max-oracle in our max-oracle gradient-descent algorithm by a projected gradient-ascent procedure, which again computes a $\delta$-maximum of the inner player's value function.

Once $\widehat{\boldsymbol{y}}$ is found at iteration $t$, one can compute optimal KKT multipliers $\lambda_1^*(\boldsymbol{x}^{(t)}, \widehat{\boldsymbol{y}}(\boldsymbol{x}^{(t)}))$, ..., $\lambda_K^*(\boldsymbol{x}^{(t)}, \widehat{\boldsymbol{y}}(\boldsymbol{x}^{(t)}))$ for the outer player's value function, either via a system of linear equations using the complementary slackness conditions and the value of the objective function at the optimal, namely $(\boldsymbol{x}^{(t)}, \widehat{\boldsymbol{y}}(\boldsymbol{x}^{(t)}))$, or by running gradient descent on the Lagrangian for the dual variables. Additionally, most algorithms solving convex programs will return $\boldsymbol{\lambda}^*(\boldsymbol{x}^{(t)}, \widehat{\boldsymbol{y}}(\boldsymbol{x}^{(t)})) = (\lambda_1^*(\boldsymbol{x}^{(t)}, \widehat{\boldsymbol{y}}(\boldsymbol{x}^{(t)})), \ldots, \lambda_K^*(\boldsymbol{x}^{(t)}, \widehat{\boldsymbol{y}}(\boldsymbol{x}^{(t)})))$ together with the optimal $\widehat{\boldsymbol{y}}(\boldsymbol{x}^{(t)})$ without incurring any additional computational expense. As a result, we assume that the optimal KKT multipliers $\boldsymbol{\lambda}^*(\boldsymbol{x}^{(t)}, \widehat{\boldsymbol{y}}(\boldsymbol{x}^{(t)}))$ associated with a solution $\widehat{\boldsymbol{y}}(\boldsymbol{x}^{(t)})$ can be computed in constant time.

Having explained our two procedures, our next task is to derive their convergence rates. It turns out that under very mild assumptions, i.e., when Assumption 3.1 holds, the outer player's value function is Lipschitz continuous in $\boldsymbol{x}$. More precisely, the objective function $f$ is $\ell_f$-Lipschitz in $\boldsymbol{x}$, where $\ell_f = \max_{(\widehat{\boldsymbol{x}}, \widehat{\boldsymbol{y}}) \in X \times Y} \|\nabla_{\boldsymbol{x}} f(\widehat{\boldsymbol{x}}, \widehat{\boldsymbol{y}})\|$.[4] As $f$ is $\ell_f$-Lipschitz continuous in $\boldsymbol{x}$, the value function is

---

[4]This max norm is well-defined since $\nabla_{\boldsymbol{x}} f$ is continuous and the constraint set is non-empty and compact.

---

**Algorithm 1** Max-Oracle Gradient Descent

---

**Inputs:** $X, Y, f, \boldsymbol{g}, \boldsymbol{\eta}, T, \boldsymbol{x}^{(0)}$
**Output:** $(\boldsymbol{x}^*, \boldsymbol{y}^*)$
 1: **for** $t = 1, \dots, T$ **do**
 2:     Find $\boldsymbol{y}^{(t-1)} \in Y$ s.t. $f(\boldsymbol{x}^{(t-1)}, \boldsymbol{y}^{(t-1)}) \geq V(\boldsymbol{x}^{(t-1)}) - \delta$ & $\boldsymbol{g}(\boldsymbol{x}^{(t-1)}, \boldsymbol{y}^{(t-1)}) \geq \boldsymbol{0}$
 3:     Set $\boldsymbol{\lambda}^{(t-1)} = \boldsymbol{\lambda}^*(\boldsymbol{x}^{(t-1)}, \boldsymbol{y}^{(t-1)})$
 4:     Set $\boldsymbol{x}^{(t)} = \Pi_X \left( \boldsymbol{x}^{(t-1)} - \eta_t \left[ \nabla_{\boldsymbol{x}} f(\boldsymbol{x}^{(t-1)}, \boldsymbol{y}^{(t-1)}) + \sum_{k=1}^{K} \lambda_k^{(t-1)} \nabla_{\boldsymbol{x}} g_k(\boldsymbol{x}^{(t-1)}, \boldsymbol{y}^{(t-1)}) \right] \right)$
 5: **end for**
 6: Find $\boldsymbol{y}^{(T)} \in Y$ s.t. $f(\boldsymbol{x}^{(T)}, \boldsymbol{y}^{(T)}) \geq V(\boldsymbol{x}^{(T)}) - \delta$ & $\boldsymbol{g}(\boldsymbol{x}^{(T)}, \boldsymbol{y}^{(T)}) \geq \boldsymbol{0}$
 7: **return** $(\boldsymbol{x}^{(T)}, \boldsymbol{y}^{(T)})$

---

also $\ell_f$-Lipschitz.[5] This fact suggests that an $(\varepsilon, \varepsilon)$-Stackelberg equilibrium should be computable in $O(\varepsilon^{-2})$ iterations by our max-oracle gradient descent algorithm (Algorithm 1), since our method is a subgradient method.

**Theorem 3.3.** *Consider a convex-concave min-max Stackelberg game, $(X, Y, f, \boldsymbol{g})$, where $X$ is convex, and $(\boldsymbol{x}^*, \boldsymbol{y}^*)$ is a $(0, \delta)$-Stackelberg equilibrium, and suppose Assumption 3.1 holds. If Algorithm 1 is run for $T$ iterations, with step sizes $\eta_t$ s.t. $\sum_{k=1}^{T} \eta_k^2 \leq \infty, \sum_{k=1}^{T} \eta_k = \infty$, and if $(\boldsymbol{x}_{\text{best}}^{(t)}, \boldsymbol{y}_{\text{best}}^{(t)}) \in \arg\min_{(\boldsymbol{x}^{(k)}, \boldsymbol{y}^{(k)}): k \in [t]} f(\boldsymbol{x}^{(k)}, \boldsymbol{y}^{(k)})$, it holds that $\lim_{T \to \infty} f(\boldsymbol{x}_{\text{best}}^{(T)}, \boldsymbol{x}_{\text{best}}^{(T)}) = f(\boldsymbol{x}^*, \boldsymbol{y}^*)$. Furthermore, for $\varepsilon \in (0, 1)$, if we choose $T \geq N_T(\varepsilon) \in O(\varepsilon^{-2})$, then there exists an iteration $T^* \leq T$ s.t. $(\boldsymbol{x}_{\text{best}}^{(T^*)}, \boldsymbol{y}_{\text{best}}^{(T^*)})$ is an $(\varepsilon, \delta)$-Stackelberg equilibrium.*

As is expected, the $O(\varepsilon^{-2})$ iteration complexity can be improved to $O(\varepsilon^{-1})$, if additionally, $f$ is strongly convex in $\boldsymbol{x}$. (See Appendix E, Theorem E.2). Combining the convergence results for our max-oracle gradient descent algorithm with convergence results for gradient descent [8], we obtain the following convergence rates for the nested gradient descent-ascent algorithm (Algorithm 2, Appendix E). We include the formal proof and statement for the case when Assumption 3.1 holds and $f$ is Lipschitz-smooth in Appendix E (Theorem E.3). The other results follow similarly.

**Theorem 3.4.** *Consider a convex-concave min-max Stackelberg game, $(X, Y, f, \boldsymbol{g})$, with $X$ and $Y$ convex. Suppose that Assumption 3.1 holds. Then, under standard assumptions on the step sizes, the iteration complexities given below hold for the computation of a $(\varepsilon, \varepsilon)$-Stackelberg equilibrium:*

| | $f$ is $\ell_{\nabla f}$-smooth | $f$ is $\ell_{\nabla f}$-smooth + $f$ Strongly Concave in $\boldsymbol{y}$ |
|---|---|---|
| *Assumption 3.1* | $O(\varepsilon^{-3})$ | $O\left(\varepsilon^{-2} \log(\varepsilon^{-1})\right)$ |
| *Assumption 3.1 + $f$ Strongly Convex in $\boldsymbol{x}$* | $O(\varepsilon^{-2})$ | $O(\varepsilon^{-1} \log(\varepsilon^{-1}))$ |

Since the value function in the convex-concave dependent setting is not guaranteed to be differentiable (see Example C.2), we cannot ensure that the objective function is Lipschitz-smooth in general. Thus, unlike previous results for the independent setting that required this latter assumption to achieve faster convergence (e.g., [58]), in our analysis of Algorithm 1, we assume only that the objective function is continuously differentiable, which leads to a more widely applicable, albeit slower, convergence rate. Note, however, that we assume Lipschitz-smoothness in our analysis of Algorithm 2, as it allows for faster convergence to the inner player's optimal strategy, but this assumption could also be done away with again, at the cost of a slower convergence rate.

## 4 An Economic Application: Fisher Markets

The Fisher market model, attributed to Irving Fisher [10], has received a great deal of attention recently, in particular by computer scientists, as its applications to fair division and mechanism design have proven useful for the design of automated markets in many online marketplaces. In this section, we argue that a competitive equilibrium in Fisher markets can be understood a Stackelberg

---

[5]A formal statement of this claim can be found in Appendix E (Lemma E.1).

equilibrium of a convex-concave min-max Stackelberg game. We then apply our first-order methods to compute these equilibria in various Fisher markets.

A **Fisher market** consists of $n$ buyers and $m$ divisible goods [10]. Each buyer $i \in [n]$ has a budget $b_i \in \mathbb{R}_+$ and a utility function $u_i : \mathbb{R}_+^m \to \mathbb{R}$. As is standard in the literature, we assume that there is one divisible unit of each good in the market [57]. An instance of a Fisher market is given by a tuple $(n, m, U, \boldsymbol{b})$, where $U = \{u_1, \dots, u_n\}$ is a set of utility functions, one per buyer, and $\boldsymbol{b} \in \mathbb{R}_+^n$ is the vector of buyer budgets. We abbreviate as $(U, \boldsymbol{b})$ when $n$ and $m$ are clear from context.

An **allocation** $\boldsymbol{X} = (\boldsymbol{x}_1, \dots, \boldsymbol{x}_n)^T \in \mathbb{R}_+^{n \times m}$ is a map from goods to buyers, represented as a matrix, s.t. $x_{ij} \geq 0$ denotes the amount of good $j \in [m]$ allocated to buyer $i \in [n]$. Goods are assigned **prices** $\boldsymbol{p} = (p_1, \dots, p_m)^T \in \mathbb{R}_+^m$. A tuple $(\boldsymbol{p}^*, \boldsymbol{X}^*)$ is said to be a **competitive (or Walrasian) equilibrium** of Fisher market $(U, \boldsymbol{b})$ if 1. buyers are utility maximizing, constrained by their budget, i.e., $\forall i \in [n], \boldsymbol{x}_i^* \in \arg\max_{\boldsymbol{x} : \boldsymbol{x} \cdot \boldsymbol{p}^* \leq b_i} u_i(\boldsymbol{x})$; and 2. the market clears, i.e., $\forall j \in [m], p_j^* > 0 \Rightarrow \sum_{i \in [n]} x_{ij}^* = 1$ and $p_j^* = 0 \Rightarrow \sum_{i \in [n]} x_{ij}^* \leq 1$.

We now formulate the problem of computing a competitive equilibrium $(\boldsymbol{p}^*, \boldsymbol{X}^*)$ of a Fisher market $(U, \boldsymbol{b})$, where $U$ is a set of continuous, concave, and homogeneous[6] utility functions, as a convex-concave min-max Stackelberg game, a perspective which has not been taken before. Fisher markets can by solved via the Eisenberg-Gale convex program [24]. Recently, Cole and Tao [13] derived a convex program, which is intimately related to the dual of the Eisenberg-Gale program [31], namely

$$\min_{\boldsymbol{p} \geq \boldsymbol{0}} \sum_{j \in [m]} p_j + \sum_{i \in [n]} b_i \log \left( \max_{\boldsymbol{x}_i \geq \boldsymbol{0} : \boldsymbol{x}_i \cdot \boldsymbol{p} \leq b_i} u_i(\boldsymbol{x}_i) \right) \quad . \tag{2}$$

Rearranging, we obtain the following convex-concave min-max Stackelberg game:

$$\min_{\boldsymbol{p} \geq \boldsymbol{0}} \max_{\boldsymbol{X} \geq \boldsymbol{0} : \boldsymbol{X}\boldsymbol{p} \leq \boldsymbol{b}} \sum_{j \in [m]} p_j + \sum_{i \in [n]} b_i \log \left( u_i(\boldsymbol{x}_i) \right) \quad . \tag{3}$$

This min-max game is played by a fictitious (Walrasian) auctioneer and a set of buyers, who effectively play as a team. The objective function in this game is then the sum of the auctioneer's welfare (i.e., the sum of the prices) and the Nash social welfare of buyers (i.e., the second summation). As the buyer's strategy set is dependent on the price vector $\boldsymbol{p}$ selected by the auctioneer, we cannot use existing first-order methods to solve this problem. However, we can use Algorithms 1 and 2.

Starting from Equation (3), define the auctioneer's value function $V(\boldsymbol{p}) = \max_{\boldsymbol{X} \geq \boldsymbol{0} : \boldsymbol{X}\boldsymbol{p} \leq \boldsymbol{b}} \sum_{j \in [m]} p_j + \sum_{i \in [n]} b_i \log \left( u_i(\boldsymbol{x}_i) \right)$, and buyer $i$'s demand set $X_i^*(\boldsymbol{p}, \boldsymbol{b}) = \arg\max_{\boldsymbol{x}_i \geq \boldsymbol{0} : \boldsymbol{x}_i \boldsymbol{p} \leq b_i} u_i(\boldsymbol{x}_i)$. Theorem 3.2 then provides the relevant subgradients so that we can run Algorithms 1 and 2, namely $\partial_{\boldsymbol{p}} V(\boldsymbol{p}) = \boldsymbol{1}_m - \sum_{i \in [n]} X_i^*(\boldsymbol{p}, \boldsymbol{b})$ and $\nabla_{\boldsymbol{x}_i} \left( \sum_{j \in [m]} p_j + \sum_{i \in [n]} b_i \log \left( u_i(\boldsymbol{x}_i) \right) \right) = \frac{b_i}{u_i(\boldsymbol{x}_i)} \nabla_{\boldsymbol{x}_i} u_i(\boldsymbol{x}_i)$, using the Minkowski sum to add set-valued quantities, where $\boldsymbol{1}_m$ is the vector of ones of size $m$.[7]

Cheung, Cole, and Devanur [12] observed that solving the dual of the Eisenberg-Gale program (Equation 2) via (sub)gradient descent [20] is equivalent to solving for a competitive equilibrium in a Fisher market using an auction-like economic price adjustment process named tâtonnement that was first proposed by Léon Walras in the 1870s [80]. The tâtonnement process increases the prices of goods that are overdemanded and decreases the prices of goods that are underdemanded. Mathematically, the (vanilla) tâtonnement process [5, 80] is defined as $\boldsymbol{p}(t) = \max \left\{ \boldsymbol{p}(t-1) + \eta_t \left( \sum_{i \in [n]} \boldsymbol{x}_i^*(\boldsymbol{p}(t), \boldsymbol{b}) - 1 \right), 0 \right\}$ for $\boldsymbol{p}(0) \in \mathbb{R}_+^m$, where $\boldsymbol{x}_i^*(\boldsymbol{p}(t), b_i) \in X_i^*(\boldsymbol{p}(t), b_i) = \arg\max_{\boldsymbol{x}_i \geq \boldsymbol{0} : \boldsymbol{x}_i \boldsymbol{p}(t) \leq b_i} u_i(\boldsymbol{x}_i)$ is the demand set of buyer $i$. The max-oracle algorithm applied to Equation (3) is then equivalent to a tâtonnemement process where the buyers report a $\delta$-utility maximizing demand. Further, we have the following corollary of Theorem 3.3.

**Corollary 4.1.** *Let $(U, \boldsymbol{b})$ be a Fisher market with equilibrium price vector $\boldsymbol{p}^*$, where $U$ is a set of continuous, concave, homogeneous, and continuously differentiable utility functions. Consider*

---

[6] A function $f : \mathbb{R}^m \to \mathbb{R}$ is said to be **homogeneous of degree** $k$ if $\forall \boldsymbol{x} \in \mathbb{R}^m, \lambda > 0, f(\lambda \boldsymbol{x}) = \lambda^k f(\boldsymbol{x})$. Unless otherwise indicated, a homogeneous function is assumed to be homogeneous of degree 1.

[7] We include detailed descriptions of the algorithms applied to Fisher markets in Appendix F.

*the entropic tâtonnement process [31].*[8] *Assume that the step sizes $\eta_t$ satisfy the usual conditions:* $\sum_{k=1}^{T} \eta_k^2 \leq \infty$ *and* $\sum_{k=1}^{T} \eta_k = \infty$. *If* $\boldsymbol{p}_{\text{best}}^{(t)} \in \arg\min_{\boldsymbol{p}^{(k)}:k\in[t]} V(\boldsymbol{p}^{(k)})$, *then* $\lim_{k\to\infty} V(\boldsymbol{p}_{\text{best}}^{(k)}) = V(\boldsymbol{p}^*)$. *Additionally, tâtonnement converges to an $\varepsilon$-competitive equilibrium in $O(\varepsilon^{-2})$ iterations.*

If we also apply the nested gradient-descent-ascent algorithm to Equation (3), we arrive at an algorithm that is arguably more descriptive of market dynamics than tâtonnement itself, as it also includes the demand-side market dynamics of buyers optimizing their demands, potentially in a decentralized manner. The nested tâtonnement algorithm essentially describes a two-step trial-and-error (i.e., tâtonnement) process, where first the buyers try to discover their optimal demand by increasing their demand for goods in proportion to the marginal utility the goods provide, and then the seller/auctioneer adjusts market prices by decreasing the prices of goods that are underdemanded and increasing the prices of goods that are overdemanded. As buyers can calculate their demands in a decentralized fashion, the nested tâtonnement algorithm offers a more complete picture of market dynamics then the classic tâtonnement process.

**Experiments**   In order to better understand, the iteration complexity of Algorithms 1 and 2 (Appendix E), we ran a series of experiments on Fisher markets with three different classes of utility functions.[9] Each utility structure endows Equation (3) with different smoothness properties, which allows us to compare the efficiency of the algorithms under varying conditions.

Let $\boldsymbol{v}_i \in \mathbb{R}^m$ be a vector of parameters for the utility function of buyer $i \in [n]$. We have the following utility function classes: Linear: $u_i(\boldsymbol{x}_i) = \sum_{j\in[m]} v_{ij} x_{ij}$, Cobb-Douglas: $u_i(\boldsymbol{x}_i) = \prod_{j\in[m]} x_{ij}^{v_{ij}}$, Leontief: $u_i(\boldsymbol{x}_i) = \min_{j\in[m]} \left\{ \frac{x_{ij}}{v_{ij}} \right\}$. Equation (3) satisfies the smoothness properties listed in Table 2 when $U$ is one of these three classes. Our goals are two-fold. First, we want to understand how the empirical convergence rates of Algorithms 1 and 2 (which, when applied to Equation (3) give rise to Algorithms 3 and 4 in Appendix F, respectively) compare to their theoretical guarantees under different utility structures. Second, we want to understand the extent to which the convergence rates of these two algorithms differ in practice. We include a more detailed description of our experimental setup in Appendix F.

Table 2: Smoothness properties satisfied by Equation (3) assuming different utility functions. Note that Assumption 3.1 does not hold for Leontief utilities, because they are not differentiable.

|              | $V$ is differentiable | Assumption 3.1 holds |
|--------------|:---------------------:|:--------------------:|
| Linear       | ×                     | ✓                    |
| Cobb-Douglas | ✓                     | ✓                    |
| Leontief     | ✓                     | ×                    |

Figure 1 describes the empirical convergence rates of Algorithms 1 and 2 for linear, Cobb-Douglas, and Leontief utilities. We observe that convergence is fastest in Fisher markets with Cobb-Douglas utilities, followed by linear, and then Leontief. We seem to obtain a tight convergence rate of $O(1/\sqrt{T})$ for linear utilities, which seems plausible, as the value function is not differentiable assuming linear utilities, and hence we are unlikely to achieve a better convergence rate. On the other hand, for Cobb-Douglas utilities, both the value and the objective function are differentiable; in fact, they are both twice continuously differentiable, making them both Lipschitz-smooth. These factors combined seem to provide a much faster convergence rate than $O(1/\sqrt{T})$.

Fisher markets with Leontief utilities, in which the objective function is not differentiable, are the hardest markets of the three for our algorithms to solve. Indeed, our theory does not even predict convergence. Still, convergence is not entirely surprising, as Cheung, Cole, and Devanur [12] have shown that buyer demand throughout tâtonnement is bounded for Leontief utilities, which means that the objective function of Equation (3) is locally Lipschitz: i.e., any subgradient computed by the algorithm will be bounded. Overall, our theory suggests that differentiability of the value function is not essential to guarantee convergence of first-order methods in convex-concave games, while our

---

[8]The entropic tâtonnement process ensures that prices remain bounded away from zero, so that excess demand is likewise bounded and Assumption 3.1 is satisfied.

[9]Our code can be found at `https://github.com/denizalp/min-max-fisher.git`.

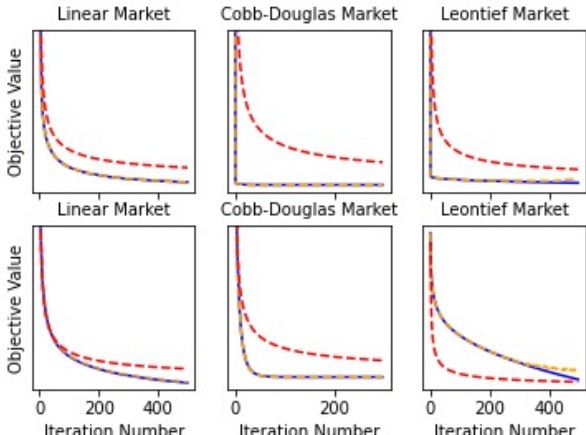

Figure 1: The first row describes the average trajectory of the value of the objective function for a randomly initialized market on each iteration of both Algorithm 3 (in blue) and Algorithm 4 (in orange) when the starting prices are high, while the second row describes the average trajectory of the objective function when starting prices are low for linear, Cobb-Douglas, and Leontief Fisher markets respectively. The dashed red line represents a convergence rate of $O(1/\sqrt{T})$, which corresponds to an iteration complexity of $O(1/\epsilon^2)$.

experiments seem to suggest that differentiability of the objective function is more important than differentiability of the value function in regards to the convergence rate.

In order to investigate whether the outputs of Algorithm 1, which uses an exact max-oracle (i.e., 0-max-oracle) are more precise than those of Algorithm 2, we solved 500 randomly initialized markets with both algorithms. We then ran a James' first-order test [2, 36] on the mean output of both algorithms to see if their difference was statistically significant. Our calculations produced $p$-values of 0.69, 0, and $1.06 \times 10^{-18}$, for Fisher markets with linear, Cobb-Douglas, and Leontief utilities, respectively. At a significance level of 0.05, these results are *not* statistically significant for linear utilities only. This result can be attributed to the fact that the value function is not differentiable in the linear case, which makes the nested gradient descent/ascent algorithm less precise.

## 5   Conclusion

In this paper, we proposed two first-order methods to solve a class of constrained convex-concave min-max optimization problems, which we call min-max Stackelberg games. As such games do not afford Nash equilibria in general, we focused on the computation of their Stackelberg equilibria. To solve for Stackelberg equilibria, we introduced a novel subdifferential envelope theorem, which formed the heart of two subgradient methods with polynomial-time iteration complexity that converge to Stackelberg equilibria. Finally, we applied our theory to the computation of competitive equilibria in Fisher markets. This application yielded a new variant of the classic tâtonnement process, where, in addition to the auctioneer iteratively adjusting prices, the buyers iteratively compute their demands. A further question of interest both for Fisher market dynamics and convex-concave min-max Stackelberg games more generally is whether gradient-descent-ascent (GDA) converges in the dependent strategy set setting as it does in the independent strategy setting [45]. GDA dynamics for Fisher markets correspond to myopic best-response dynamics (see, for example, [52]). We would expect such a result to be of computational as well as economic interest.

## Acknowledgments and Disclosure of Funding

We would like to thank Dustin Morrill for his feedback on an early draft of this paper. This work was partially supported by NSF Grant CMMI-1761546.

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
