# A  Pseudo-Games and Generalized Nash Equilibria

**Pseudo-games**, or **abstract economies** [4], are optimization problems that are closely related to min-max Stackelberg games, but which are technically not games, as noted by Facchinei and Kanzow [26, 27], because each player's strategy set is not fixed at the outset (i.e., before they have to make a decision), but instead depends on the other players' choices. [39]. In this appendix, we formally define two-player, zero-sum pseudo-games,[10] and discuss how they differ from min-max Stackelberg games. We also define the equilibrium concept *par excellence* of pseudo-games, namely **generalized Nash equilibrium**, and juxtapose its definition with vanilla Nash equilibrium.

A two-player, zero-sum **pseudo-game** comprises two players, with respective payoff functions $-f(\boldsymbol{x}, \boldsymbol{y})$ and $f(\boldsymbol{x}, \boldsymbol{y})$, and respective strategy spaces given by the correspondences $\mathcal{X}: Y \rightrightarrows X$ and $\mathcal{Y}: X \rightrightarrows Y$, i.e., set valued mappings that depend on the choice the other player takes. More specifically, $\mathcal{X}(\boldsymbol{y}) = \{\boldsymbol{x} \in X \mid \forall k \in [K], g_k(\boldsymbol{x}, \boldsymbol{y}) \geq 0\}$ and $\mathcal{Y}(\boldsymbol{x}) = \{\boldsymbol{y} \in Y \mid \forall k \in [K], g_k(\boldsymbol{x}, \boldsymbol{y}) \geq 0\}$. We denote a pseudo-game by $(X, Y, f, \boldsymbol{g})$.

Pseudo-games are closely related to min-max Stackelberg games, as they both comprise agents with the same objectives and the same space of feasible strategy profiles, namely $\{(\boldsymbol{x}, \boldsymbol{y}) \in X \times Y \mid \forall k \in [K], g_k(\boldsymbol{x}, \boldsymbol{y}) \geq 0\}$. However, whereas play proceeds sequentially in Stackelberg games, so that inner player chooses a strategy $\boldsymbol{y} \in Y$ s.t. $\boldsymbol{g}(\widehat{\boldsymbol{x}}, \boldsymbol{y}) \geq 0$ *after* the outer player chooses $\widehat{\boldsymbol{x}} \in X$, in pseudo-games players choose their strategies simultaneously. In other words, the space of feasible strategy profiles is an emergent feature of a pseudo-game; it depends on the players' choices, which are made simultaneously, yet must somehow still be feasible, even though one player's choice could render another player's choice infeasible.

A **generalized Nash equilibrium (GNE)** of pseudo-game $(X, Y, f, \boldsymbol{g})$ is a strategy profile $(\boldsymbol{x}^*, \boldsymbol{y}^*) \in X \times Y$ s.t. $\boldsymbol{g}(\boldsymbol{x}^*, \boldsymbol{y}^*) \geq 0$ such that

$$\max_{\boldsymbol{y} \in Y: \boldsymbol{g}(\boldsymbol{x}^*, \boldsymbol{y}) \geq 0} f(\boldsymbol{x}^*, \boldsymbol{y}) \leq f(\boldsymbol{x}^*, \boldsymbol{y}^*) \leq \min_{\boldsymbol{x} \in X: \boldsymbol{g}(\boldsymbol{x}, \boldsymbol{y}^*) \geq 0} f(\boldsymbol{x}, \boldsymbol{y}^*) \ . \tag{4}$$

That is, at a GNE the players choose best responses to the other players' strategies from within the space of strategies defined by the other players' choices. Under Assumption 3.1, a GNE is guaranteed to exist for any pseudo-game $(X, Y, f, \boldsymbol{g})$. A Nash equilibrium, however, is not guaranteed to exist.

A **Nash equilibrium (NE)** of a min-max Stackelberg game $(X, Y, f, \boldsymbol{g})$ is a tuple $(\boldsymbol{x}^*, \boldsymbol{y}^*) \in X \times Y$ such that

$$\max_{\boldsymbol{y} \in Y} \min_{\boldsymbol{x} \in X: \boldsymbol{g}(\boldsymbol{x}, \boldsymbol{y}) \geq 0} f(\boldsymbol{x}, \boldsymbol{y}) \leq f(\boldsymbol{x}^*, \boldsymbol{y}^*) \leq \min_{\boldsymbol{x} \in X} \max_{\boldsymbol{y} \in Y: \boldsymbol{g}(\boldsymbol{x}, \boldsymbol{y}) \geq 0} f(\boldsymbol{x}, \boldsymbol{y}) \ . \tag{5}$$

That is, at a NE the players best respond to one another's choices, under the assumption that the other player is doing the same: i.e., best responding to their choice.

We summarize what is known about min-max Stackelberg games vs. pseudo-games, and Stackelberg vs. generalized Nash (and vanilla Nash) equilibria, as follows. Min-max Stackelberg games are proper games, while pseudo-games are not well-defined games. Stackelberg min-max games are played sequentially, while pseudo-games are played simultaneously. When Nash equilibria exist in min-max Stackelberg games, they coincide with Stackelberg equilibria. Additionally, Nash equilibria coincide with generalized Nash equilibria in min-max games with independent strategy sets. There exist no polynomial-time computation guaantees for generalized Nash equilibria in pseudo-games, even in the two-player, zero-sum setting, while we provide polynomial-time computation guarantees for Stackelberg equilibria in min-max Stackelberg games.

# B  Omitted Proofs Section 2

**Proposition B.1.** *Consider a min-max Stackelberg game $(X, Y, f, \boldsymbol{g})$. Then if Assumption 3.1 holds, the outer player's value function, $V(\boldsymbol{x}) = \max_{\boldsymbol{y} \in Y: \boldsymbol{g}(\boldsymbol{x}, \boldsymbol{y}) \geq \boldsymbol{0}} f(\boldsymbol{x}, \boldsymbol{y})$, is continuous and convex.*

*Proof.* By Berge's maximum theorem [7], the outer player's value function is continuous.

---

[10]We refer the reader to Facchinei and Kanzow's [26, 27] survey on pseudo-games for a more detailed exposition, beyond two-player, zero-sum pseudo-games.

Define $\mathcal{L} : Y \times \mathbb{R}_+^K \times X \to \mathbb{R}$ s.t. $\mathcal{L}(\boldsymbol{y}, \boldsymbol{\lambda}; \boldsymbol{x}) = f(\boldsymbol{x}, \boldsymbol{y}) + \sum_{k=1}^K \lambda_k g_k(\boldsymbol{x}, \boldsymbol{y})$ and $h : Y \times X \to \mathbb{R}$ s.t. $h(\boldsymbol{y}; \boldsymbol{x}) = \min_{\boldsymbol{\lambda} \in \mathbb{R}_+^K} \mathcal{L}(\boldsymbol{y}, \boldsymbol{\lambda}; \boldsymbol{x})$. Since Slater's condition is satisfied under Assumption 3.1, the KKT theorem [44] applies, which means that for all $\boldsymbol{x} \in X$ and $\boldsymbol{y} \in Y$, the optimal KKT multipliers $\boldsymbol{\lambda}^*$ exist and are well-defined: i.e, there exists $c \in \mathbb{R}_+$ such that $\forall k \in [K], 0 \le \lambda_k^* \le c < \infty$. We can thus re-express the outer player's value function $V(\boldsymbol{x})$ as the following Lagrangian saddle-point problems:

$$V(\boldsymbol{x}) = \max_{\boldsymbol{y} \in Y : \boldsymbol{g}(\boldsymbol{x}, \boldsymbol{y}) \ge \boldsymbol{0}} f(\boldsymbol{x}, \boldsymbol{y}) = \max_{\boldsymbol{y} \in Y} \min_{\boldsymbol{\lambda} \in \mathbb{R}_+^K} \mathcal{L}(\boldsymbol{y}, \boldsymbol{\lambda}; \boldsymbol{x}) = \max_{\boldsymbol{y} \in Y} \min_{\boldsymbol{\lambda} \in [0, c]^K} \mathcal{L}(\boldsymbol{y}, \boldsymbol{\lambda}; \boldsymbol{x}) = \max_{\boldsymbol{y} \in Y} h(\boldsymbol{y}; \boldsymbol{x}) \ . \tag{6}$$

Notice that $\mathcal{L}$ is concave in $\boldsymbol{y}$, convex in $\boldsymbol{\lambda}$, and convex in $\boldsymbol{x}$. Since $h$ is the min-projection of $\mathcal{L}$ w.r.t. $\boldsymbol{\lambda}$ onto the compact set $[0, c]^K$, $h$ must be convex in $\boldsymbol{x}$ ([65], Proposition 2.22). Further, since $V$ is the maximum of convex functions over a compact set, i.e., $\max_{\boldsymbol{y} \in Y} h(\boldsymbol{y}; \boldsymbol{x})$, $V$ must be convex ([65], Theorem 5.5). $\qquad\square$

**Proposition B.2.** *Given any min-max Stackelberg game $(X, Y, f, \boldsymbol{g})$, a Stackelberg equilibrium always exists and its value is unique.*

*Proof of Proposition B.2.* By Berge's maximum theorem [7], the outer player's value function $V(\boldsymbol{x}) = \max_{\boldsymbol{y} \in Y : \boldsymbol{g}(\boldsymbol{x}, \boldsymbol{y}) \ge \boldsymbol{0}} f(\boldsymbol{x}, \boldsymbol{y})$ is continuous, and the inner player's solution correspondence $Y^*(\boldsymbol{x}) = \arg \max_{\boldsymbol{y} \in Y : \boldsymbol{g}(\boldsymbol{x}, \boldsymbol{y}) \ge \boldsymbol{0}} f(\boldsymbol{x}, \boldsymbol{y})$ is non-empty, for all $\boldsymbol{x} \in X$. Since $V$ is continuous and $X$ is compact and non-empty, by the extreme value theorem [62], there exists a minimizer $\boldsymbol{x}^* \in X$ of $V$. Hence $(\boldsymbol{x}^*, \boldsymbol{y}^*(\boldsymbol{x}^*))$, with $\boldsymbol{y}^*(\boldsymbol{x}^*) \in Y^*(\boldsymbol{x}^*)$, is a Stackelberg equilibrium of $(X, Y, f, \boldsymbol{g})$.

Let $(\boldsymbol{x}_1, \boldsymbol{y}_1)$ and $(\boldsymbol{x}_2, \boldsymbol{y}_2)$ be two different Stackelberg equilibria. WLOG, suppose $f(\boldsymbol{x}_1, \boldsymbol{y}_1) > f(\boldsymbol{x}_2, \boldsymbol{y}_2)$, so that $V(\boldsymbol{x}_1) = f(\boldsymbol{x}_1, \boldsymbol{y}_1) > f(\boldsymbol{x}_2, \boldsymbol{y}_2) = V(\boldsymbol{x}_2)$, where the first and last equality follow from the definition of Stackelberg equilibrium. But then $(\boldsymbol{x}_1, \boldsymbol{y}_1)$ cannot be a Stackelberg equilibrium, since $\boldsymbol{x}_1$ is not a minimizer of the outer player's value function. $\qquad\square$

**Proposition B.3.** *The set of Stackelberg equilibria of any min-max Stackelberg game $(X, Y, f, \boldsymbol{g})$ is non-empty, compact, and convex.*

*Proof.* By Proposition B.2, the set of Stackelberg equilibria of any min-max Stackelberg game is non-empty. Additionally, under Assumption 3.1, we have that for all $\boldsymbol{x} \in \mathcal{X}$, $f(\boldsymbol{x}, \cdot)$ is concave, and $\{\boldsymbol{y} \in Y \mid \boldsymbol{g}(\boldsymbol{x}, \boldsymbol{y}) \ge \boldsymbol{0}\}$ is convex. Hence, by Theorem 2.6 of Rockafeller [65], the set of solutions $Y^*(\boldsymbol{x}) = \arg \max_{\boldsymbol{y} \in Y : \boldsymbol{g}(\boldsymbol{x}, \boldsymbol{y}) \ge \boldsymbol{0}} f(\boldsymbol{x}, \boldsymbol{y})$ is compact- and convex- valued. Similarly, by Proposition B.1, under Assumption 3.1, $\max_{\boldsymbol{y} \in Y : \boldsymbol{g}(\boldsymbol{x}, \boldsymbol{y}) \ge \boldsymbol{0}} f(\boldsymbol{x}, \boldsymbol{y})$ is continuous and convex. Hence, the set of solutions $X^* = \arg \min_{\boldsymbol{x} \in X} \max_{\boldsymbol{y} \in Y : \boldsymbol{g}(\boldsymbol{x}, \boldsymbol{y}) \ge \boldsymbol{0}} f(\boldsymbol{x}, \boldsymbol{y})$ is compact- and convex-valued. Since the composition of two compact-convex-valued correspondences is again compact-convex-valued (Proposition 5.52 of Rockafeller [65]), we conclude that the set of Stackelberg equilibria, namely $X^*(Y^*)$, is compact and convex. $\qquad\square$

# C   Envelope Theorem

Danskin's theorem [17] offers insights into optimization problems of the form: $\max_{\boldsymbol{y} \in Y} f(\boldsymbol{x}, \boldsymbol{y})$, where $Y \subset \mathbb{R}^m$ is compact and non-empty. Among other things, Danskin's theorem allows us to compute the gradient of the objective function of this optimization problem with respect to $\boldsymbol{x}$.

**Theorem C.1** (Danskin's Theorem). *Consider an optimization problem of the form: $\max_{\boldsymbol{y} \in Y} f(\boldsymbol{x}, \boldsymbol{y})$, where $Y \subset \mathbb{R}^m$ is compact and non-empty. Suppose that $Y$ is convex and that $f$ is concave in $\boldsymbol{y}$. Let $V(\boldsymbol{x}) = \max_{\boldsymbol{y} \in Y} f(\boldsymbol{x}, \boldsymbol{y})$ and $Y^*(\boldsymbol{x}) = \arg \max_{\boldsymbol{y} \in Y} f(\boldsymbol{x}, \boldsymbol{y})$. Then $V$ is differentiable at $\widehat{\boldsymbol{x}}$, if the solution correspondence $Y^*(\widehat{\boldsymbol{x}})$ is a singleton: i.e., $Y^*(\widehat{\boldsymbol{x}}) = \{\boldsymbol{y}^*(\widehat{\boldsymbol{x}})\}$. Additionally, the gradient at $\widehat{\boldsymbol{x}}$ is given by $V'(\widehat{\boldsymbol{x}}) = \nabla_{\boldsymbol{x}} f(\widehat{\boldsymbol{x}}, \boldsymbol{y}^*(\widehat{\boldsymbol{x}}))$.*

Unfortunately, Danskin's theorem does not hold when the set $Y$ is replaced by a correspondence, which occurs in min-max Stackelberg games: i.e., when the inner problem is $\max_{\boldsymbol{y} \in Y : \boldsymbol{g}(\boldsymbol{x}, \boldsymbol{y}) \ge \boldsymbol{0}} f(\boldsymbol{x}, \boldsymbol{y})$.

**Example C.2** (Danskin's theorem does not apply to min-max Stackelberg games)**.** *Consider the optimization problem:*

$$\max_{y \in \mathbb{R}: y+x \geq 0} -y^2 + y + 2x + 2 \ . \tag{7}$$

*The solution to this problem is unique, given any $x \in X$, meaning the solution correspondence $Y^*(x)$ is singleton-valued. We denote this unique solution by $y^*(x)$. After solving, we find that*

$$y^*(x) = \begin{cases} 1/2 & \text{if } x \geq -1/2 \\ -x & \text{if } x < -1/2 \end{cases} \tag{8}$$

*The value function $V(x) = \max_{y \in \mathbb{R}: y+x \geq 0} -y^2 + y + 2x + 2$ is then given by:*

$$V(x) = f(x, y^*(x)) \tag{9}$$

$$= -y^*(x)^2 + y^*(x) + 2x + 2 \tag{10}$$

$$= \begin{cases} -1/4 + 1/2 + 2x + 2 & \text{if } x \geq -1/2 \\ -x^2 - x + 2x + 2 & \text{if } x < -1/2 \end{cases} \tag{11}$$

$$= \begin{cases} 9/4 + 2x & \text{if } x \geq -1/2 \\ -x^2 + x + 2 & \text{if } x < -1/2 \end{cases} \tag{12}$$

*The derivative of this value function is:*

$$\frac{\partial V}{\partial x} = \begin{cases} 2 & \text{if } x \geq -1/2 \\ 1 - 2x & \text{if } x < -1/2 \end{cases} \tag{13}$$

*However, the derivative predicted by Danskin's theorem is 2. Hence, Danskin's theorem does not hold when the constraints are parameterized, i.e., when the problem is of the form $\min_{\boldsymbol{y} \in Y: \boldsymbol{g}(\boldsymbol{x}, \boldsymbol{y})} f(\boldsymbol{x}, \boldsymbol{y})$ rather than $\min_{\boldsymbol{y} \in Y} f(\boldsymbol{x}, \boldsymbol{y})$, where $X \subset \mathbb{R}^n$, $Y \subset \mathbb{R}^m$, and for all $k \in [K]$, $g_k : X \times Y \to \mathbb{R}$ are continuous.*

***N.B.*** *For simplicity, we do not assume the constraint set is compact in this example. Compactness of the constraint set can be used to guarantee existence of a solution, but as a solution to this particular problem always exists, we can do away with this assumption.*

The following theorem, due to Milgrom and Segal [50], generalizes Danskin's theorem to handle parameterized constraints:

**Theorem C.3** (Envelope Theorem [50])**.** *Consider the maximization problem*

$$V(\boldsymbol{x}) = \max_{\boldsymbol{y} \in Y} f(\boldsymbol{x}, \boldsymbol{y}), \text{ subject to } g_k(\boldsymbol{x}, \boldsymbol{y}) \geq 0, \text{ for all } k = 1, \dots, K \ , \tag{14}$$

*where $Y \subseteq \mathbb{R}^m$.*

*Define the solution correspondence $Y^*(\boldsymbol{x}) = \arg\max_{\boldsymbol{y} \in X: \boldsymbol{g}(\boldsymbol{x}, \boldsymbol{y}) \geq \boldsymbol{0}} f(\boldsymbol{x}, \boldsymbol{y})$. If Assumption 3.1 holds, then the value function $V$ is absolutely continuous, and at any point $\widehat{\boldsymbol{x}} \in X$ where $V$ is differentiable:*

$$\nabla_{\boldsymbol{x}} V(\widehat{\boldsymbol{x}}) = \nabla_{\boldsymbol{x}} L(\boldsymbol{y}^*(\widehat{\boldsymbol{x}}), \boldsymbol{\lambda}^*(\widehat{\boldsymbol{x}}, \boldsymbol{y}^*(\widehat{\boldsymbol{x}}))), \widehat{\boldsymbol{x}}) = \nabla_{\boldsymbol{x}} f(\widehat{\boldsymbol{x}}, \boldsymbol{y}^*(\widehat{\boldsymbol{x}})) + \sum_{k=1}^{K} \lambda_k^*(\widehat{\boldsymbol{x}}, \boldsymbol{y}^*(\widehat{\boldsymbol{x}})) \nabla_{\boldsymbol{x}} g_k(\widehat{\boldsymbol{x}}, \boldsymbol{y}^*(\widehat{\boldsymbol{x}})) \ , \tag{15}$$

*where $\boldsymbol{\lambda}^*(\widehat{\boldsymbol{x}}, \boldsymbol{y}^*(\widehat{\boldsymbol{x}})) = (\lambda_1(\widehat{\boldsymbol{x}}, \boldsymbol{y}^*(\widehat{\boldsymbol{x}})), \dots, \lambda_K^*(\widehat{\boldsymbol{x}}, \boldsymbol{y}^*(\widehat{\boldsymbol{x}})))^T \in \Lambda^*(\widehat{\boldsymbol{x}}, \boldsymbol{y}^*(\widehat{\boldsymbol{x}}))$ are the KKT multipliers associated associated with $\boldsymbol{y}^*(\widehat{\boldsymbol{x}}) \in Y^*(\widehat{\boldsymbol{x}})$.*

# D   Omitted Subdifferential Envelope Theorem Proof (Section 3)

*Proof of Theorem 3.2.* As usual, let $V(\boldsymbol{x}) = \max_{\boldsymbol{y} \in Y: \boldsymbol{g}(\boldsymbol{x}, \boldsymbol{y}) \geq 0} f(\boldsymbol{x}, \boldsymbol{y})$. First, note that Proposition B.1 $V$ is subdifferentiable as it is convex [9]. Reformulating the problem as a Lagrangian saddle point problem, for all $\widehat{\boldsymbol{x}} \in X$, it holds that:

$$V(\widehat{\boldsymbol{x}}) = \max_{\boldsymbol{y} \in Y: \boldsymbol{g}(\widehat{\boldsymbol{x}}, \boldsymbol{y}) \geq 0} f(\widehat{\boldsymbol{x}}, \boldsymbol{y}) \tag{16}$$

$$= \max_{\boldsymbol{y} \in Y} \min_{\boldsymbol{\lambda} \in \mathbb{R}_{++}^K} \left\{ f(\widehat{\boldsymbol{x}}, \boldsymbol{y}) + \sum_{k=1}^{K} \lambda_k g_k(\widehat{\boldsymbol{x}}, \boldsymbol{y}) \right\} \tag{17}$$

Since $f$ is continuous, $Y$ is compact, and $g_1, \ldots, g_K$ are continuous, for all $\widehat{\boldsymbol{x}} \in X$, there exists $\boldsymbol{y}^*(\widehat{\boldsymbol{x}}) \in \arg\max_{\boldsymbol{y} \in Y : \boldsymbol{g}(\widehat{\boldsymbol{x}}, \boldsymbol{y}) \geq \boldsymbol{0}} f(\widehat{\boldsymbol{x}}, \boldsymbol{y})$. Furthermore, as Assumption 3.1 ensures that an interior solution exists, the Karush-Kuhn-Tucker Theorem [44] applies, so for all $\widehat{\boldsymbol{x}} \in X$ and any associated $\boldsymbol{y}^*(\widehat{\boldsymbol{x}})$, there exists $\boldsymbol{\lambda}(\widehat{\boldsymbol{x}}, \boldsymbol{y}^*(\widehat{\boldsymbol{x}})) \in \mathbb{R}^K$ that solves Equation (17).

Define the solution correspondence $Y^*(\boldsymbol{x}) = \arg\max_{\boldsymbol{y} \in Y : \boldsymbol{g}(\boldsymbol{x}, \boldsymbol{y}) \geq \boldsymbol{0}} f(\boldsymbol{x}, \boldsymbol{y})$, and let $\Lambda^*(\boldsymbol{x}, \boldsymbol{y}) = \arg\min_{\boldsymbol{\lambda} \in \mathbb{R}_+^K} \left\{ f(\boldsymbol{x}, \boldsymbol{y}) + \sum_{k=1}^K \lambda_k g_k(\boldsymbol{x}, \boldsymbol{y}) \right\}$. We can then re-express the value function at $\widehat{\boldsymbol{x}}$ as:

$$V(\widehat{\boldsymbol{x}}) = f(\widehat{\boldsymbol{x}}, \boldsymbol{y}^*(\widehat{\boldsymbol{x}})) + \sum_{k=1}^K \lambda_k^*(\widehat{\boldsymbol{x}}, \boldsymbol{y}^*(\widehat{\boldsymbol{x}})) g_k(\widehat{\boldsymbol{x}}, \boldsymbol{y}^*(\widehat{\boldsymbol{x}})), \quad \forall \boldsymbol{y}^*(\widehat{\boldsymbol{x}}) \in Y^*(\widehat{\boldsymbol{x}}), \lambda_k^*(\widehat{\boldsymbol{x}}, \boldsymbol{y}^*(\widehat{\boldsymbol{x}})) \in \Lambda^*(\widehat{\boldsymbol{x}}, \boldsymbol{y}^*(\widehat{\boldsymbol{x}})) \ .$$

Equivalently, we can take the maximum over $\boldsymbol{y}^*$'s and $\boldsymbol{\lambda}^*$'s to obtain:

$$V(\widehat{\boldsymbol{x}}) = \max_{\boldsymbol{y}^*(\widehat{\boldsymbol{x}}) \in Y^*(\widehat{\boldsymbol{x}})} \max_{\lambda_k^*(\widehat{\boldsymbol{x}}, \boldsymbol{y}^*(\widehat{\boldsymbol{x}})) \in \Lambda^*(\widehat{\boldsymbol{x}}, \boldsymbol{y}^*(\widehat{\boldsymbol{x}}))} \left\{ f(\widehat{\boldsymbol{x}}, \boldsymbol{y}) + \sum_{k=1}^K \lambda_k^*(\widehat{\boldsymbol{x}}, \boldsymbol{y}^*(\widehat{\boldsymbol{x}})) g_k(\widehat{\boldsymbol{x}}, \boldsymbol{y}) \right\} \ .$$

Note that for fixed $\boldsymbol{y}^*(\widehat{\boldsymbol{x}}) \in Y^*(\widehat{\boldsymbol{x}})$ and corresponding fixed $\lambda_k^*(\widehat{\boldsymbol{x}}, \boldsymbol{y}^*(\widehat{\boldsymbol{x}})) \in \Lambda^*(\widehat{\boldsymbol{x}}, \boldsymbol{y}^*(\widehat{\boldsymbol{x}}))$, $f(\widehat{\boldsymbol{x}}, \boldsymbol{y}) + \sum_{k=1}^K \lambda_k^*(\widehat{\boldsymbol{x}}, \boldsymbol{y}^*(\widehat{\boldsymbol{x}})) g_k(\widehat{\boldsymbol{x}}, \boldsymbol{y}^*(\widehat{\boldsymbol{x}}))$ is differentiable, since $f, g_1, \ldots, g_K$ are differentiable.

Additionally, recall the pointwise maximum subdifferential property,[11] i.e., if $f(\boldsymbol{x}) = \max_{\alpha \in \mathcal{A}} f_\alpha(\boldsymbol{x})$ for a family of functions $\{f_\alpha\}_{\alpha \in \mathcal{A}}$, then $\partial_{\boldsymbol{x}} f(\boldsymbol{a}) = \mathrm{conv}\left( \bigcup_{\alpha \in \mathcal{A}} \{\partial_{\boldsymbol{x}} f_{\alpha \in \mathcal{A}}(\boldsymbol{a}) \mid f_\alpha(\boldsymbol{a}) = f(\boldsymbol{x})\} \right)$, which then gives:

$$\partial_{\boldsymbol{x}} V(\widehat{\boldsymbol{x}}) = \partial_{\boldsymbol{x}} \left( \max_{\boldsymbol{y}^*(\widehat{\boldsymbol{x}}) \in Y^*(\widehat{\boldsymbol{x}})} \max_{\lambda_k^*(\widehat{\boldsymbol{x}}, \boldsymbol{y}^*(\widehat{\boldsymbol{x}})) \in \Lambda^*(\widehat{\boldsymbol{x}}, \boldsymbol{y}^*(\widehat{\boldsymbol{x}}))} \left\{ f(\widehat{\boldsymbol{x}}, \boldsymbol{y}^*(\widehat{\boldsymbol{x}})) + \sum_{k=1}^K \lambda_k^*(\widehat{\boldsymbol{x}}, \boldsymbol{y}^*(\widehat{\boldsymbol{x}})) g_k(\widehat{\boldsymbol{x}}, \boldsymbol{y}^*(\widehat{\boldsymbol{x}})) \right\} \right)$$
(18)

$$= \mathrm{conv}\left( \bigcup_{\boldsymbol{y}^*(\widehat{\boldsymbol{x}}) \in Y^*(\widehat{\boldsymbol{x}})} \bigcup_{\lambda_k^*(\widehat{\boldsymbol{x}}, \boldsymbol{y}^*(\widehat{\boldsymbol{x}})) \in \Lambda^*(\widehat{\boldsymbol{x}}, \boldsymbol{y}^*(\widehat{\boldsymbol{x}}))} \partial_{\boldsymbol{x}} \left\{ f(\widehat{\boldsymbol{x}}, \boldsymbol{y}^*(\widehat{\boldsymbol{x}})) + \sum_{k=1}^K \lambda_k^*(\widehat{\boldsymbol{x}}, \boldsymbol{y}^*(\widehat{\boldsymbol{x}})) g_k(\widehat{\boldsymbol{x}}, \boldsymbol{y}^*(\widehat{\boldsymbol{x}})) \right\} \right)$$
(19)

$$= \mathrm{conv}\left( \bigcup_{\boldsymbol{y}^*(\widehat{\boldsymbol{x}}) \in Y^*(\widehat{\boldsymbol{x}})} \bigcup_{\lambda_k^*(\widehat{\boldsymbol{x}}, \boldsymbol{y}^*(\widehat{\boldsymbol{x}})) \in \Lambda^*(\widehat{\boldsymbol{x}}, \boldsymbol{y}^*(\widehat{\boldsymbol{x}}))} \left\{ \nabla_{\boldsymbol{x}} f(\widehat{\boldsymbol{x}}, \boldsymbol{y}^*(\widehat{\boldsymbol{x}})) + \sum_{k=1}^K \lambda_k^*(\widehat{\boldsymbol{x}}, \boldsymbol{y}^*(\widehat{\boldsymbol{x}})) \nabla_{\boldsymbol{x}} g_k(\widehat{\boldsymbol{x}}, \boldsymbol{y}^*(\widehat{\boldsymbol{x}})) \right\} \right) \ .$$
(20)

$\square$

# E   Pseudo-Codes and Convergence Results for Section 3

**Lemma E.1** (Lipschitz Objective Function implies Lipschitz Value Function). *Let $f : X \times Y$ be a continuous function, where $X \subset \mathbb{R}^n$ and $Y \subset \mathbb{R}^m$. If $\nabla_{\boldsymbol{x}} f$ is continuous in $(\boldsymbol{x}, \boldsymbol{y})$, $X, Y$ are compact and non-empty, and for all $k \in [K]$, $g_k$ is continuous , then $V(\boldsymbol{x}) = \max_{\boldsymbol{y} \in Y : \boldsymbol{g}(\boldsymbol{x}, \boldsymbol{y}) \geq \boldsymbol{0}} f(\boldsymbol{x}, \boldsymbol{y})$ is $\ell_f$-Lipschitz continuous: i.e., $\|V(\boldsymbol{x}_1) - V(\boldsymbol{x}_2)\| \leq \ell_f \|\boldsymbol{x}_1 - \boldsymbol{x}_2\|$, with $\ell_f = \max_{(\widehat{\boldsymbol{x}}, \widehat{\boldsymbol{y}}) \in X \times Y} \|\nabla_{\boldsymbol{x}} f(\widehat{\boldsymbol{x}}, \widehat{\boldsymbol{y}})\|$.*

*Proof of Lemma E.1.* Let $\ell_f = \max_{(\widehat{\boldsymbol{x}}, \widehat{\boldsymbol{y}}) \in X \times Y} \|\nabla_{\boldsymbol{x}} f(\widehat{\boldsymbol{x}}, \widehat{\boldsymbol{y}})\|$. Because $\nabla_{\boldsymbol{x}} f$ is continuous, $f$ is $\ell_f$-Lipschitz continuous in $\boldsymbol{x}$: i.e., $\forall \boldsymbol{x}_1, \boldsymbol{x}_2 \in X, \boldsymbol{y} \in Y, \|f(\boldsymbol{x}_1, \boldsymbol{y}) - f(\boldsymbol{x}_2, \boldsymbol{y})\| \leq \ell_f \|\boldsymbol{x}_1 - \boldsymbol{x}_2\|$.

Fix $\boldsymbol{x}_1, \boldsymbol{x}_2 \in X$. Then, for all $\boldsymbol{y} \in Y$, we have:

$$f(\boldsymbol{x}_1, \boldsymbol{y}) \leq f(\boldsymbol{x}_1, \boldsymbol{y}) - f(\boldsymbol{x}_2, \boldsymbol{y}) + f(\boldsymbol{x}_2, \boldsymbol{y}) \tag{21}$$
$$\leq \ell_f \|\boldsymbol{x}_1 - \boldsymbol{x}_2\| + f(\boldsymbol{x}_2, \boldsymbol{y}) \ . \tag{22}$$

---

[11] See, for example, [9].

---

**Algorithm 2** Nested Gradient Descent

**Inputs:** $X, Y, f, \boldsymbol{g}, \boldsymbol{\eta^x}, \boldsymbol{\eta^y}, T_{\boldsymbol{x}}, T_{\boldsymbol{y}}, \boldsymbol{x}^{(0)}, \boldsymbol{y}^{(0)}$
**Output:** $\boldsymbol{x}^*, \boldsymbol{y}^*$

1: **for** $t = 1, \ldots, T_{\boldsymbol{x}}$ **do**
2:      $\boldsymbol{y}^{(t-1)} = \boldsymbol{y}^{(0)}$
3:      **for** $s = 1, \ldots, T_{\boldsymbol{y}}$ **do**
4:          $\boldsymbol{y}^{(t-1)} = \Pi_{\{\boldsymbol{y} \in Y : \boldsymbol{g}(\boldsymbol{x}^{(t-1)}, \boldsymbol{y}) \geq \boldsymbol{0}\}} \left( \boldsymbol{y}^{(t-1)} + \eta_s^{\boldsymbol{y}} \left[ \nabla_{\boldsymbol{y}} f(\boldsymbol{x}^{(t-1)}, \boldsymbol{y}^{(t-1)}) \right] \right)$
5:      **end for**
6:      Set $\boldsymbol{\lambda}^{(t-1)} = \boldsymbol{\lambda}^*(\boldsymbol{x}^{(t-1)}, \boldsymbol{y}^{(t-1)})$
7:      Set $\boldsymbol{x}^{(t)} = \Pi_X \left( \boldsymbol{x}^{(t-1)} - \eta_t^{\boldsymbol{x}} \left[ \nabla_{\boldsymbol{x}} f(\boldsymbol{x}^{(t-1)}, \boldsymbol{y}^{(t-1)}) + \sum_{k=1}^K \lambda_k^{(t-1)} \nabla_{\boldsymbol{x}} g_k(\boldsymbol{x}^{(t-1)}, \boldsymbol{y}^{(t-1)}) \right] \right)$
8: **end for**
9: $\boldsymbol{y}^{(T)} = \boldsymbol{y}^{(0)}$
10: **for** $s = 1, \ldots, T_{\boldsymbol{y}}$ **do**
11:      $\boldsymbol{y}^{(T)} = \Pi_{\{\boldsymbol{y} \in Y : \boldsymbol{g}(\boldsymbol{x}^{(T)}, \boldsymbol{y}) \geq \boldsymbol{0}\}} \left( \boldsymbol{y}^{(T)} + \eta_s^{\boldsymbol{y}} \nabla_{\boldsymbol{y}} f(\boldsymbol{x}^{(T)}, \boldsymbol{y}^{(T)}) \right]$
12: **end for**
13: **return** $(\boldsymbol{x}^{(T)}, \boldsymbol{y}^{(T)})$

---

Taking the max over the $\boldsymbol{y}$'s on both sides (which is guaranteed to exist by the continuity of $f$, and of the $g_k$'s, and the fact that $Y$ is non-empty and compact) yields:

$$\max_{\boldsymbol{y} \in Y : \boldsymbol{g}(\boldsymbol{x}_1, \boldsymbol{y}) \geq 0} f(\boldsymbol{x}_1, \boldsymbol{y}) \leq \ell_f \|\boldsymbol{x}_1 - \boldsymbol{x}_2\| + \max_{\boldsymbol{y} \in Y : \boldsymbol{g}(\boldsymbol{x}_2, \boldsymbol{y}) \geq 0} f(\boldsymbol{x}_2, \boldsymbol{y}) \tag{23}$$

$$V(\boldsymbol{x}_1) \leq \ell_f \|\boldsymbol{x}_1 - \boldsymbol{x}_2\| + V(\boldsymbol{x}_2) \tag{24}$$

$$V(\boldsymbol{x}_1) - V(\boldsymbol{x}_2) \leq \ell_f \|\boldsymbol{x}_1 - \boldsymbol{x}_2\| . \tag{25}$$

Since this inequality holds for arbitrary $\boldsymbol{x}_1, \boldsymbol{x}_2 \in X$, we also have:

$$V(\boldsymbol{x}_2) - V(\boldsymbol{x}_1) \leq \ell_f \|\boldsymbol{x}_1 - \boldsymbol{x}_2\| . \tag{26}$$

Combining these two inequalities, it holds that: $\|V(\boldsymbol{x}_1) - V(\boldsymbol{x}_2)\| \leq \ell_f \|\boldsymbol{x}_1 - \boldsymbol{x}_2\|$. $\qquad\square$

*Proof of Theorem 3.3.* By our subdifferential envelope theorem (Theorem 3.2), we have:

$$\nabla_{\boldsymbol{x}} f(\boldsymbol{x}^{(t-1)}, \boldsymbol{y}^{(t-1)}) + \sum_{k=1}^K \lambda_k^{(t-1)} \nabla_{\boldsymbol{x}} g_k(\boldsymbol{x}^{(t-1)}, \boldsymbol{y}^{(t-1)}) \in \partial_{\boldsymbol{x}} V(\boldsymbol{x}^{(t-1)}) \tag{27}$$

$$= \partial_{\boldsymbol{x}} \max_{\boldsymbol{y} \in Y : \boldsymbol{g}(\boldsymbol{x}^{(t-1)}, \boldsymbol{y}) \geq 0} f(\boldsymbol{x}^{(t-1)}, \boldsymbol{y}) . \tag{28}$$

For notational clarity, let $\boldsymbol{h}(t-1) = \nabla_{\boldsymbol{x}} f(\boldsymbol{x}^{(t-1)}, \boldsymbol{y}^{(t-1)}) + \sum_{k=1}^K \lambda_k^{(t-1)} \nabla_{\boldsymbol{x}} g_k(\boldsymbol{x}^{(t-1)}, \boldsymbol{y}^{(t-1)})$.
Suppose that $\boldsymbol{x}^* \in \arg\min_{\boldsymbol{x} \in X} \max_{\boldsymbol{y} \in Y : \boldsymbol{g}(\boldsymbol{x}, \boldsymbol{y}) \geq \boldsymbol{0}} f(\boldsymbol{x}, \boldsymbol{y})$. Then:

$$\left\| \boldsymbol{x}^{(T)} - \boldsymbol{x}^* \right\|^2 \tag{29}$$

$$= \left\| \Pi_X \left( \boldsymbol{x}^{(T-1)} - \eta_T \boldsymbol{h}(T-1) \right) - \Pi_X \left( \boldsymbol{x}^* \right) \right\|^2 \tag{30}$$

$$\leq \left\| \boldsymbol{x}^{(T-1)} - \eta_T \boldsymbol{h}(T-1) - \boldsymbol{x}^* \right\|^2 \tag{31}$$

$$= \left\| \boldsymbol{x}^{(T-1)} - \boldsymbol{x}^* \right\|^2 - 2\eta_T \left\langle \boldsymbol{h}(T-1), \left( \boldsymbol{x}^{(T-1)} - \boldsymbol{x}^* \right) \right\rangle + \eta_T^2 \|\boldsymbol{h}(T-1)\|^2 \tag{32}$$

$$\leq \left\| \boldsymbol{x}^{(T-1)} - \boldsymbol{x}^* \right\|^2 - 2\eta_T \left( f(\boldsymbol{x}^{(T-1)}, \boldsymbol{y}^{(T-1)}) - f(\boldsymbol{x}^*, \boldsymbol{y}^{(T-1)}) \right) + \eta_T^2 \|\boldsymbol{h}(T-1)\|^2 , \tag{33}$$

where the first line follows from the subgradient descent rule and the fact that $\boldsymbol{x}^* \in X$; the second, because the project operator is a non-expansion; the third, by the definition of the norm; and the

fourth, by the definition of the subgradient, i.e., $\langle \boldsymbol{h}(t-1), \left(\boldsymbol{x}^{(t-1)} - \boldsymbol{x}^*\right)\rangle \geq f(\boldsymbol{x}^{(t-1)}, \boldsymbol{y}^{(t-1)}) - f(\boldsymbol{x}^*, \boldsymbol{y}^{(t-1)})$. Applying this inequality recursively, we obtain:

$$\left\|\boldsymbol{x}^{(T)} - \boldsymbol{x}^*\right\|^2 \leq \left\|\boldsymbol{x}^{(0)} - \boldsymbol{x}^*\right\|^2 - \sum_{t=1}^{T} 2\eta_t \left(f(\boldsymbol{x}^{(t-1)}, \boldsymbol{y}^{(t-1)}) - f(\boldsymbol{x}^*, \boldsymbol{y}^{(t-1)})\right) + \sum_{t=1}^{T} \eta_t^2 \left\|\boldsymbol{h}(k-1)\right\|^2 \quad . \tag{34}$$

Since $\left\|\boldsymbol{x}^{(t)} - \boldsymbol{x}^*\right\| \geq 0$, we have:

$$2\sum_{t=1}^{T} \eta_t \left(f(\boldsymbol{x}^{(t-1)}, \boldsymbol{y}^{(t-1)}) - f(\boldsymbol{x}^*, \boldsymbol{y}^{(t-1)})\right) \leq \left\|\boldsymbol{x}^{(0)} - \boldsymbol{x}^*\right\|^2 + \sum_{t=1}^{T} \eta_t^2 \left\|\boldsymbol{h}(t-1)\right\|^2 \quad . \tag{35}$$

Let $(\boldsymbol{x}_{\text{best}}^{(t)}, \boldsymbol{y}_{\text{best}}^{(t)}) \in \arg\min_{(\boldsymbol{x}^{(k)}, \boldsymbol{y}^{(k)}):k\in[t]} f(\boldsymbol{x}^{(k)}, \boldsymbol{y}^{(k)})$. Then:

$$\sum_{t=1}^{T} \eta_t \left(f(\boldsymbol{x}^{(t-1)}, \boldsymbol{y}^{(t-1)}) - f(\boldsymbol{x}^*, \boldsymbol{y}^{(t-1)})\right) \tag{36}$$

$$\geq \sum_{t=1}^{T} \eta_t \left(f(\boldsymbol{x}^{(t-1)}, \boldsymbol{y}^{(t-1)}) - \max_{\boldsymbol{y}\in Y: \boldsymbol{g}(\boldsymbol{x}^*, \boldsymbol{y})\geq \boldsymbol{0}} f(\boldsymbol{x}^*, \boldsymbol{y})\right) \tag{37}$$

$$\geq \left(\sum_{t=1}^{T} \eta_t\right) \min_{t\in[T]} \left(f(\boldsymbol{x}^{(t-1)}, \boldsymbol{y}^{(t-1)}) - \max_{\boldsymbol{y}\in Y: \boldsymbol{g}(\boldsymbol{x}^*, \boldsymbol{y})\geq \boldsymbol{0}} f(\boldsymbol{x}^*, \boldsymbol{y})\right) \tag{38}$$

$$= \left(\sum_{t=1}^{T} \eta_t\right) \left(f(\boldsymbol{x}_{\text{best}}^{(T)}, \boldsymbol{x}_{\text{best}}^{(T)}) - \max_{\boldsymbol{y}\in Y: \boldsymbol{g}(\boldsymbol{x}^*, \boldsymbol{y})\geq \boldsymbol{0}} f(\boldsymbol{x}^*, \boldsymbol{y})\right) \tag{39}$$

Hence, we get the following bound:

$$f(\boldsymbol{x}_{\text{best}}^{(T)}, \boldsymbol{y}_{\text{best}}^{(T)}) - \max_{\boldsymbol{y}\in Y: \boldsymbol{g}(\boldsymbol{x}^*, \boldsymbol{y})\geq \boldsymbol{0}} f(\boldsymbol{x}^*, \boldsymbol{y}) \leq \frac{\left\|\boldsymbol{x}^{(0)} - \boldsymbol{x}^*\right\|^2 + \sum_{t=1}^{T} \eta_t^2 \|\boldsymbol{h}(t-1)\|^2}{2\left(\sum_{t=1}^{T} \eta_t\right)} \tag{40}$$

Now since $f$ is $\ell_f$-Lipschitz with $\ell_f = \max_{(\widehat{\boldsymbol{x}}, \widehat{\boldsymbol{y}})\in X\times Y} \|\nabla_{\boldsymbol{x}} f(\widehat{\boldsymbol{x}}, \widehat{\boldsymbol{y}})\|$, all the subgradients are bounded: i.e., for all $k\in\mathbb{N}$, $\|\boldsymbol{h}(k-1)\| \leq \ell_f$. So:

$$f(\boldsymbol{x}_{\text{best}}^{(T)}, \boldsymbol{x}_{\text{best}}^{(T)}) - \max_{\boldsymbol{y}\in Y: \boldsymbol{g}(\boldsymbol{x}^*, \boldsymbol{y})\geq \boldsymbol{0}} f(\boldsymbol{x}^*, \boldsymbol{y}) \leq \frac{\left\|\boldsymbol{x}^{(0)} - \boldsymbol{x}^*\right\|^2 + \ell_f^2 \sum_{t=1}^{T} \eta_t^2}{2\left(\sum_{t=1}^{T} \eta_t\right)} \tag{41}$$

Equivalently,

$$f(\boldsymbol{x}_{\text{best}}^{(T)}, \boldsymbol{x}_{\text{best}}^{(T)}) - \min_{\boldsymbol{x}\in X} \max_{\boldsymbol{y}\in Y: \boldsymbol{g}(\boldsymbol{x}, \boldsymbol{y})\geq \boldsymbol{0}} f(\boldsymbol{x}, \boldsymbol{y}) \leq \frac{\left\|\boldsymbol{x}^{(0)} - \boldsymbol{x}^*\right\|^2 + \ell_f^2 \sum_{t=1}^{T} \eta_t^2}{2\left(\sum_{t=1}^{T} \eta_t\right)} \tag{42}$$

Recall the assumptions that the step sizes are square-summable but not summable, namely $\sum_{k=1}^{T} \eta_k^2 \leq \infty$ and $\sum_{k=1}^{T} \eta_k = \infty$. Now as $T\to\infty$, Equation (42) becomes:

$$\lim_{k\to\infty} f(\boldsymbol{x}_{\text{best}}^{(k)}, \boldsymbol{y}^{(k)}) \leq \min_{\boldsymbol{x}\in X} \max_{\boldsymbol{y}\in Y: \boldsymbol{g}(\boldsymbol{x}, \boldsymbol{y})\geq \boldsymbol{0}} f(\boldsymbol{x}, \boldsymbol{y}) \quad . \tag{43}$$

We have thus proven the first inequality of the two that define an $(0, \delta)$-Stackelberg equilibrium.

The second inequality follows by construction, as for all $k\in\mathbb{N}$, the max oracle returns $\boldsymbol{y}_{\text{best}}^{(k)}$ that satisfies $f(\boldsymbol{x}_{\text{best}}^{(k)}, \boldsymbol{y}_{\text{best}}^{(k)}) \geq \max_{\boldsymbol{y}\in Y: \boldsymbol{g}(\boldsymbol{x}, \boldsymbol{y})\geq \boldsymbol{0}} f(\boldsymbol{x}_{\text{best}}^{(k)}, \boldsymbol{y}) - \delta$. Thus, as $k\to\infty$, the best iterate converges to a $(0, \delta)$-Stackelberg equilibrium.

Additionally, setting $\eta_t = \frac{\left\|\boldsymbol{x}^{(0)} - \boldsymbol{x}^*\right\|}{\ell_f \sqrt{T}}$, we find that for all $t \in [T]$,

$$f(\boldsymbol{x}_{\text{best}}^{(T)}, \boldsymbol{x}_{\text{best}}^{(T)}) - \min_{\boldsymbol{x} \in X} \max_{\boldsymbol{y} \in Y : \boldsymbol{g}(\boldsymbol{x}, \boldsymbol{y}) \geq 0} f(\boldsymbol{x}, \boldsymbol{y}) \leq \frac{\ell_f \left\|\boldsymbol{x}^{(0)} - \boldsymbol{x}^*\right\|^2}{\sqrt{T}} \tag{44}$$

Likewise, setting $\varepsilon \leq \frac{\ell_f \left\|\boldsymbol{x}^{(0)} - \boldsymbol{x}^*\right\|^2}{\sqrt{T}}$, we obtain $T \leq \frac{\ell_f^2 \left\|\boldsymbol{x}^{(0)} - \boldsymbol{x}^*\right\|^4}{\varepsilon^2}$, and thus $N_T(\varepsilon) \in O(\varepsilon^{-2})$.

Finally, applying the same logic as above to infer the second inequality in the Stackelberg equilibrium definition, the best iterate converges to a $(\varepsilon, \delta)$-Stackelberg equilibrium in $O(\varepsilon^{-2})$ iterations. $\qquad\square$

**Theorem E.2.** *Suppose Algorithm 1 is run on a convex-concave min-max Stackelberg game given by $(X, Y, f, \boldsymbol{g})$ where $X$ is convex. Suppose further that Assumption 3.1 holds, and that $f$ is $\mu$-strongly convex in $\boldsymbol{x}$, i.e., $\forall \boldsymbol{x}_1, \boldsymbol{x}_2 \in X, \boldsymbol{y} \in Y$,*

$$f(\boldsymbol{x}_1, \boldsymbol{y}) \geq f(\boldsymbol{x}_2, \boldsymbol{y}) + \langle \boldsymbol{h}, (\boldsymbol{x}_1 - \boldsymbol{x}_2) \rangle + \frac{\mu}{2} \|\boldsymbol{x}_1 - \boldsymbol{x}_2\|^2 \quad, \tag{45}$$

*where $\boldsymbol{h} \in \partial_{\boldsymbol{x}} f(\boldsymbol{x}_2, \boldsymbol{y})$. Then, if $(\boldsymbol{x}_{\text{best}}^{(t)}, \boldsymbol{y}_{\text{best}}^{(t)}) \in \arg\min_{(\boldsymbol{x}^{(k)}, \boldsymbol{y}^{(k)}) : k \in [t]} f(\boldsymbol{x}^{(k)}, \boldsymbol{y}^{(k)})$, for $\varepsilon \in (0, 1)$, and $\eta_t = \frac{2}{\mu(t+1)}$, if we choose $T \geq N_T(\varepsilon) \in O(\varepsilon^{-1})$, then there exists an iteration $T^* \leq T$ s.t. $(\boldsymbol{x}_{\text{best}}^{(T^*)}, \boldsymbol{y}_{\text{best}}^{(T^*)})$ is an $(\varepsilon, \delta)$-Stackelberg equilibrium.*

*Proof of Theorem E.2.* By our subdifferential envelope theorem (Theorem 3.2), we have:

$$\nabla_{\boldsymbol{x}} f(\boldsymbol{x}^{(t-1)}, \boldsymbol{y}^{(t-1)}) + \sum_{k=1}^{K} \lambda_k^{(t-1)} \nabla_{\boldsymbol{x}} g_k(\boldsymbol{x}^{(t-1)}, \boldsymbol{y}^{(t-1)}) \in \partial_{\boldsymbol{x}} V(\boldsymbol{x}^{(t-1)}) \tag{46}$$

$$= \partial_{\boldsymbol{x}} \max_{\boldsymbol{y} \in Y : \boldsymbol{g}(\boldsymbol{x}^{(t-1)}, \boldsymbol{y}) \geq 0} f(\boldsymbol{x}^{(t-1)}, \boldsymbol{y}) \ . \tag{47}$$

For notational clarity, let $\boldsymbol{h}(t-1) = \nabla_{\boldsymbol{x}} f(\boldsymbol{x}^{(t-1)}, \boldsymbol{y}^{(t-1)}) + \sum_{k=1}^{K} \lambda_k^{(t-1)} \nabla_{\boldsymbol{x}} g_k(\boldsymbol{x}^{(t-1)}, \boldsymbol{y}^{(t-1)})$. Suppose that $\boldsymbol{x}^* \in \arg\min_{\boldsymbol{x} \in X} \max_{\boldsymbol{y} \in Y : \boldsymbol{g}(\boldsymbol{x}, \boldsymbol{y})} f(\boldsymbol{x}, \boldsymbol{y})$. Then, for all $t \in \mathbb{N}$ s.t. $t \geq 1$, we have:

$$\left\|\boldsymbol{x}^{(t)} - \boldsymbol{x}^*\right\|^2 \tag{48}$$

$$= \left\|\Pi_X \left(\boldsymbol{x}^{(t-1)} - \eta_t \boldsymbol{h}(t-1)\right) - \Pi_X \left(\boldsymbol{x}^*\right)\right\|^2 \tag{49}$$

$$\leq \left\|\boldsymbol{x}^{(t-1)} - \eta_t \boldsymbol{h}(t-1) - \boldsymbol{x}^*\right\|^2 \tag{50}$$

$$= \left\|\boldsymbol{x}^{(t-1)} - \boldsymbol{x}^*\right\|^2 - 2\eta_t \left\langle \boldsymbol{h}(t-1), \left(\boldsymbol{x}^{(t-1)} - \boldsymbol{x}^*\right)\right\rangle + \eta_t^2 \|\boldsymbol{h}(t-1)\|^2 \tag{51}$$

$$\leq \left\|\boldsymbol{x}^{(t-1)} - \boldsymbol{x}^*\right\|^2 - 2\eta_t \left[\frac{\mu}{2} \left\|\boldsymbol{x}^{(t-1)} - \boldsymbol{x}^*\right\|^2 + f(\boldsymbol{x}^{(t-1)}, \boldsymbol{y}^{(t)}) - f(\boldsymbol{x}^*, \boldsymbol{y}^{(t)})\right] + \eta_t^2 \|\boldsymbol{h}(t-1)\|^2 \tag{52}$$

$$= \left\|\boldsymbol{x}^{(t-1)} - \boldsymbol{x}^*\right\|^2 - \eta_t \mu \left\|\boldsymbol{x}^{(t-1)} - \boldsymbol{x}^*\right\|^2 - 2\eta_t \left(f(\boldsymbol{x}^{(t-1)}, \boldsymbol{y}^{(t-1)}) - f(\boldsymbol{x}^*, \boldsymbol{y}^{(t-1)})\right) + \eta_t^2 \|\boldsymbol{h}(t-1)\|^2 \tag{53}$$

$$= (1 - \eta_t \mu) \left\|\boldsymbol{x}^{(t-1)} - \boldsymbol{x}^*\right\|^2 - 2\eta_t \left(f(\boldsymbol{x}^{(t-1)}, \boldsymbol{y}^{(t-1)}) - f(\boldsymbol{x}^*, \boldsymbol{y}^{(t-1)})\right) + \eta_t^2 \|\boldsymbol{h}(t-1)\|^2 \tag{54}$$

where the first line follows from the subgradient descent rule and the fact that $\boldsymbol{x}^* \in X$; the second, because the project operator is a non-expansion; the third, by the definition of the norm; the fourth, by the definition of strong convexity; and the last two via straightforward algebra.

Re-organizing Equation (54), yields:

$$f(\boldsymbol{x}^{(t-1)}, \boldsymbol{y}^{(t-1)}) - f(\boldsymbol{x}^*, \boldsymbol{y}^{(t-1)}) \tag{55}$$

$$\leq \frac{1 - \eta_t \mu}{2\eta_t} \left\|\boldsymbol{x}^{(t-1)} - \boldsymbol{x}^*\right\|^2 - \frac{1}{2\eta_t} \left\|\boldsymbol{x}^{(t)} - \boldsymbol{x}^*\right\|^2 + \frac{\eta_t}{2} \|\boldsymbol{h}(t-1)\|^2 \ . \tag{56}$$

Next, setting $\eta_t = \frac{2}{\mu(t+1)}$, we get:

$$f(\boldsymbol{x}^{(t-1)}, \boldsymbol{y}^{(t-1)}) - f(\boldsymbol{x}^*, \boldsymbol{y}^{(t-1)}) \tag{57}$$

$$\leq \frac{\mu(t-1)}{4} \left\| \boldsymbol{x}^{(t-1)} - \boldsymbol{x}^* \right\|^2 - \frac{\mu(t+1)}{4} \left\| \boldsymbol{x}^{(t)} - \boldsymbol{x}^* \right\|^2 + \frac{1}{\mu(t+1)} \| \boldsymbol{h}(t-1) \|^2 \quad . \tag{58}$$

Multiplying both sides by $t$, we now have:

$$t\left( f(\boldsymbol{x}^{(t-1)}, \boldsymbol{y}^{(t-1)}) - f(\boldsymbol{x}^*, \boldsymbol{y}^{(t-1)}) \right) \tag{59}$$

$$\leq \frac{\mu t(t-1)}{4} \left\| \boldsymbol{x}^{(t-1)} - \boldsymbol{x}^* \right\|^2 - \frac{\mu t(t+1)}{4} \left\| \boldsymbol{x}^{(t)} - \boldsymbol{x}^* \right\|^2 + \frac{1}{\mu} \| \boldsymbol{h}(t-1) \|^2 \quad . \tag{60}$$

Summing up across all iterations on both sides:

$$\sum_{t=0}^{T} t\left( f(\boldsymbol{x}^{(t)}, \boldsymbol{y}^{(t)}) - f(\boldsymbol{x}^*, \boldsymbol{y}^{(t)}) \right) \tag{61}$$

$$\leq \sum_{t=0}^{T} \frac{\mu t(t-1)}{4} \left\| \boldsymbol{x}^{(t-1)} - \boldsymbol{x}^* \right\|^2 - \sum_{t=0}^{T} \frac{\mu t(t+1)}{4} \left\| \boldsymbol{x}^{(t)} - \boldsymbol{x}^* \right\|^2 + \sum_{t=0}^{T} \frac{1}{\mu} \| \boldsymbol{h}(t-1) \|^2 \tag{62}$$

$$= \sum_{t=0}^{T} \frac{\mu t(t-1)}{4} \left\| \boldsymbol{x}^{(t-1)} - \boldsymbol{x}^* \right\|^2 - \sum_{t=1}^{T+1} \frac{\mu(t-1)t}{4} \left\| \boldsymbol{x}^{(t-1)} - \boldsymbol{x}^* \right\|^2 + \sum_{t=0}^{T} \frac{1}{\mu} \| \boldsymbol{h}(t-1) \|^2 \tag{63}$$

$$= -\frac{\mu T(T+1)}{4} \left\| \boldsymbol{x}^{(t)} - \boldsymbol{x}^* \right\|^2 + \sum_{t=0}^{T} \frac{1}{\mu} \| \boldsymbol{h}(t-1) \|^2 \tag{64}$$

$$\leq \sum_{t=0}^{T} \frac{1}{\mu} \| \boldsymbol{h}(t-1) \|^2 \tag{65}$$

$$\leq \frac{T}{\mu} \ell_f \quad , \tag{66}$$

where the last line holds because $f$ is $\ell_f$-Lipschitz with $\ell_f = \max_{(\widehat{\boldsymbol{x}}, \widehat{\boldsymbol{y}}) \in X \times Y} \| \nabla_{\boldsymbol{x}} f(\widehat{\boldsymbol{x}}, \widehat{\boldsymbol{y}}) \|$, which implies that all the subgradients are bounded: i.e., for all $k \in \mathbb{N}, \| \boldsymbol{h}(k-1) \| \leq \ell_f$.

Let $(\boldsymbol{x}_{\text{best}}^{(t)}, \boldsymbol{y}_{\text{best}}^{(t)}) \in \arg\min_{(\boldsymbol{x}^{(k)}, \boldsymbol{y}^{(k)}): k \in [t]} f(\boldsymbol{x}^{(k)}, \boldsymbol{y}^{(k)})$. Then:

$$\sum_{t=0}^{T} t\left( f(\boldsymbol{x}^{(t)}, \boldsymbol{y}^{(t)}) - f(\boldsymbol{x}^*, \boldsymbol{y}^{(t)}) \right) \leq \frac{T}{\mu} \ell_f \tag{67}$$

$$\sum_{t=0}^{T} t\left( f(\boldsymbol{x}^{(t)}, \boldsymbol{y}^{(t)}) - \max_{\boldsymbol{y} \in Y: \boldsymbol{g}(\boldsymbol{x}^*, \boldsymbol{y}) \geq \boldsymbol{0}} f(\boldsymbol{x}^*, \boldsymbol{y}) \right) \leq \frac{T}{\mu} \ell_f \tag{68}$$

$$\left( \sum_{t=0}^{T} t \right) \min_{t \in [T]} \left( f(\boldsymbol{x}^{(t)}, \boldsymbol{y}^{(t)}) - \max_{\boldsymbol{y} \in Y: \boldsymbol{g}(\boldsymbol{x}^*, \boldsymbol{y}) \geq \boldsymbol{0}} f(\boldsymbol{x}^*, \boldsymbol{y}) \right) \leq \frac{T}{\mu} \ell_f \tag{69}$$

$$\left( \sum_{t=0}^{T} t \right) \left( f(\boldsymbol{x}_{\text{best}}^{(T)}, \boldsymbol{y}_{\text{best}}^{(T)}) - \max_{\boldsymbol{y} \in Y: \boldsymbol{g}(\boldsymbol{x}^*, \boldsymbol{y}) \geq \boldsymbol{0}} f(\boldsymbol{x}^*, \boldsymbol{y}) \right) \leq \frac{T}{\mu} \ell_f \tag{70}$$

$$\tag{71}$$

$$\left( \frac{(T+1)T}{2} \right) \left( f(\boldsymbol{x}_{\text{best}}^{(T)}, \boldsymbol{y}_{\text{best}}^{(T)}) - \max_{\boldsymbol{y} \in Y: \boldsymbol{g}(\boldsymbol{x}^*, \boldsymbol{y}) \geq \boldsymbol{0}} f(\boldsymbol{x}^*, \boldsymbol{y}) \right) \leq \frac{T}{\mu} \ell_f \tag{72}$$

$$f(\boldsymbol{x}_{\text{best}}^{(T)}, \boldsymbol{y}_{\text{best}}^{(T)}) - \max_{\boldsymbol{y} \in Y: \boldsymbol{g}(\boldsymbol{x}^*, \boldsymbol{y}) \geq \boldsymbol{0}} f(\boldsymbol{x}^*, \boldsymbol{y}) \leq \frac{2\ell_f}{\mu(T+1)} \quad . \tag{73}$$

That is, as the number of iterations increases, the best iterate converges to a $(0, \delta)$-Stackelberg equilibrium. Likewise, by the same logic we applied at the end of the proof as Theorem 3.3, the best iterate converges to a $(\varepsilon, \delta)$-Stackelberg equilibrium in $O(\varepsilon^{-1})$ iterations. $\qquad \square$

We now present a theorem which covers one of the cases given in Theorem 3.4. The proofs of the theorems that cover the other cases are similar to the proof below. We note that gradient ascent converges in $O(\varepsilon^{-1})$ iterations to a $\varepsilon$-maximum for a Lipschitz-smooth objective, and in $O(\log(\varepsilon))$ iterations to a $\varepsilon$-maximum for a Lipschitz-smooth and strongly-concave objective function [8].

**Theorem E.3.** *Suppose Algorithm 2 is run on a convex-concave min-max Stackelberg game given by $(X, Y, f, \boldsymbol{g})$, where $X$ and $Y$ are convex. Suppose further that Assumption 3.1 holds and that $f$ is $\ell_{\nabla f}$-smooth, i.e., $\forall (\boldsymbol{x}_1, \boldsymbol{y}_1), (\boldsymbol{x}_2, \boldsymbol{y}_2) \in X \times Y$,*

$$\|\nabla f(\boldsymbol{x}_1, \boldsymbol{y}_1) - \nabla f(\boldsymbol{x}_2, \boldsymbol{y}_2)\| \leq \ell_{\nabla f} \|(\boldsymbol{x}_1, \boldsymbol{y}_1) - (\boldsymbol{x}_2, \boldsymbol{y}_2)\| \quad . \tag{74}$$

*Let $(\boldsymbol{x}_{\text{best}}^{(t)}, \boldsymbol{y}_{\text{best}}^{(t)}) \in \arg\min_{(\boldsymbol{x}^{(k)}, \boldsymbol{y}^{(k)}):k\in[t]} f(\boldsymbol{x}^{(k)}, \boldsymbol{y}^{(k)})$. For $\varepsilon \in (0,1)$, if we choose $T_{\boldsymbol{x}}$ and $T_{\boldsymbol{y}}$ s.t. $T_{\boldsymbol{x}} \geq N_{T_{\boldsymbol{x}}}(\varepsilon) \in O(\varepsilon^{-2})$ and $T_{\boldsymbol{x}} \geq N_{T_{\boldsymbol{y}}}(\varepsilon) \in O(\varepsilon^{-1})$, then there exists an iteration $T^* \leq T_{\boldsymbol{x}} T_{\boldsymbol{y}} = O(\varepsilon^{-3})$ s.t. $(\boldsymbol{x}_{\text{best}}^{(T^*)}, \boldsymbol{y}_{\text{best}}^{(T^*)})$ is an $(\varepsilon, \varepsilon)$-Stackelberg equilibrium.*

*Proof of Theorem 3.3.* Since $f$ is $\ell_f$-smooth, it is well known that, for each outer iterate $\boldsymbol{x}^{(t)}$, the inner gradient descent procedure returns an $\varepsilon$-maximum $\boldsymbol{y}_\epsilon^*$ of $f(\boldsymbol{x}^{(t)}, \boldsymbol{y})$ s.t. $\boldsymbol{y}_\epsilon^* \in Y$ and $\boldsymbol{g}(\boldsymbol{x}^{(t)}, \boldsymbol{y}_\epsilon^*) \geq \boldsymbol{0}$, in $O(\varepsilon^{-2})$ iterations [8]. Combining the iteration complexity of the outer and inner loops using this result and Theorem 3.3, we obtain an iteration complexity of $O(\varepsilon^{-2})O(\varepsilon^{-1}) = O(\varepsilon^{-3})$. $\qquad\square$

# F  An Economic Application: Details

Our experimental goals were two-fold. First, we sought to understand the empirical convergence rate of our algorithms in different Fisher markets, in which the objective function in Equation (3) satisfies different smoothness properties. Second, we wanted to understand how the behavior of our two algorithms, max-oracle and nested gradient descent, differ in terms of the accuracy of the Stackelberg equilibria they find.

To answer these questions, we ran multiple experiments, each time recording the prices and allocations computed by Algorithm 1, with an *exact* max-oracle, and by Algorithm 2, with nested gradient ascent, during each iteration $t$ of the main (outer) loop. For each run of each algorithm on each market with each set of initial conditions, we then computed the objective function's value for the iterates, i.e., $f(\boldsymbol{x}^{(t)}, \boldsymbol{y}^{(t)})$, which we plot in Figure 1.

**Hyperparameters**   We randomly initialized 500 different linear, Cobb-Douglas, Leontief Fisher markets, each with 5 buyers and 8 goods. Buyer $i$'s budget $b_i$ was drawn randomly from a uniform distribution ranging from 100 to 1000 (i.e., $U[100, 1000]$), while each buyer $i$'s valuation for good $j$, $v_{ij}$, was drawn randomly from $U[5, 15]$. We ran both algorithms for 500, 300, and 700 iterations[12] for linear, Cobb-Douglas, and Leontief Fisher markets, respectively. We started both algorithms from two sets of initial conditions, one with high prices (drawn randomly $U[50, 55]$), and a second, with low prices (drawn randomly from $U[5, 15]$). We opted for a learning rate of 5 for both algorithms, after manual hyper-parameter tuning, and picked a decay rate of $t^{-1/2}$, based on our theory, so that $\eta_1 = 5, \eta_2 = 3.54, \eta_3 = 2.89, \eta_4 = 2.5, \eta_5 = 2.24, \ldots$.

**Programming Languages, Packages, and Licensing**   We ran our experiments in Python 3.7 [77], using NumPy [34], Pandas [73], and CVXPY [21]. Figure 1 was graphed using Matplotlib [37]. To run the first order James test, we imported the data generated by our Python code into R [63]. Our R script manipulated the data using the Tidyverse package [81], and obtained the desired $p$-values using the STests package [36].

R as a package is licensed under GPL-2 | GPL-3. Python software and documentation are licensed under the PSF License Agreement. Numpy is distributed under a liberal BSD license. Pandas is distributed under a new BSD license. Matplotlib only uses BSD compatible code, and its license is based on the PSF license. CVXPY is licensed under an APACHE license. Tidyverse is distributed under an MIT license.

---

[12]In Algorithm 3, $T \in \{500, 300, 700\}$, while in Algorithm 4, $T_{\boldsymbol{p}} \in \{500, 300, 700\}$.

**Implementation Details** In our execution of Algorithm 1 for linear, Cobb-Douglas, and Leontief Fisher markets, we used an exact Max-Oracle, since there is a closed-form solution for the demand correspondence in these markets [49].

In our execution of Algorithm 2, in order to project each computed allocation onto the consumers' budget set, i.e., $\{X \in \mathbb{R}_+^{n \times m} \mid Xp \leq b\}$, we used the alternating projection algorithm [8] for convex sets, and alternatively projected onto the sets $\mathbb{R}_+^{n \times m}$ and $\{X \in \mathbb{R}^{n \times m} \mid Xp \leq b\}$.

**Computational Resources** Our experiments were run on MacOS machine with 8GB RAM and an Apple M1 chip, and took about 2 hours to run. Only CPU resources were used.

**Code Repository** The data our experiments generated, as well as the code used to produce our visualizations and run the statistical tests, can be found in our code repository.

### F.1 Fisher Market Algorithms

---

**Algorithm 3** $\delta$-Approximate Tâtonnement for Fisher Markets

---

**Inputs:** $U, b, \eta, T, p^{(0)}, \delta$
**Output:** $(X^*, p^*)$
1: **for** $t = 1, \ldots, T$ **do**
2:      For all $i \in [n]$, find $x_i^{(t)}$ s.t. $u_i(x_i^{(t)}) \geq \max_{x_i : x_i \cdot p^{(t-1)} \leq b_i} u_i(x_i) - \delta$ & $x_i^{(t)} \cdot p^{(t-1)} \leq b_i$
3:      Set $p^{(t)} = \max \left\{ p^{(t-1)} - \eta_t (1 - \sum_{i \in [n]} x_i^{(t)}), 0 \right\}$
4: **end for**
5: **return** $(X^{(T)}, p^{(T)})$

---

---

**Algorithm 4** $\delta$-Approximate Nested Tâtonnement for Fisher Markets

---

**Inputs:** $U, b, \eta^p, \eta^X T_p, T_X, p^{(0)}$
**Output:** $(X^*, p^*)$
1: **for** $t = 1, \ldots, T_p$ **do**
2:      **for** $s = 1, \ldots, T_X$ **do**
3:          For all $i \in [n]$, $x_i^{(t)} = \Pi_{\left\{ x : x \cdot p^{(t-1)} \leq b_i \right\}} \left( x_i^{(t)} + \eta_s^X \frac{b_i}{u_i(x_i^{(t)})} \nabla_{x_i} u_i(x_i^{(t)}) \right)$
4:      **end for**
5:      Set $p^{(t)} = \max \left\{ p^{(t-1)} - \eta_t^p (1 - \sum_{i \in [n]} x_i^{(t)}), 0 \right\}$
6: **end for**
7: **return** $(X^{(T)}, p^{(T)})$

---

## G   Additional Related Work

Much progress has been made recently in solving min-max games (with independent strategy sets), both in the convex-concave case and in non-convex-concave case. For the former case, when $f$ is $\mu_x$-strongly-convex in $x$ and $\mu_y$-strongly-concave in $y$, Tseng [76], Yurii Nesterov [83], and Gidel et al. [30] proposed variational inequality methods, and Mokhtari, Ozdaglar, and Pattathil [51], gradient-descent-ascent (GDA)-based methods, all of which compute a solution in $\tilde{O}(\mu_y + \mu_x)$ iterations. These upper bounds were recently complemented by the lower bound of $\tilde{\Omega}(\sqrt{\mu_y \mu_x})$, shown by Ibrahim et al. [38] and Zhang, Hong, and Zhang [84]. Subsequently, Lin, Jin, and Jordan [46] and Alkousa et al. [3] analyzed algorithms that converge in $\tilde{O}(\sqrt{\mu_y \mu_x})$ and $\tilde{O}(\min \{\mu_x \sqrt{\mu_y}, \mu_y \sqrt{\mu_x}\})$ iterations, respectively.

For the special case where $f$ is $\mu_x$-strongly convex in $x$ and linear in $y$, Juditsky, Nemirovski, et al. [42], Hamedani and Aybat [33], and Zhao [86] all present methods that converge to an $\varepsilon$-approximate solution in $O(\sqrt{\mu_x/\varepsilon})$ iterations. When the strong concavity or linearity assumptions of $f$ on $y$ are dropped, and $f$ is assumed to be $\mu_x$-strongly-convex in $x$ but only concave in $y$, Thekumparampil

et al. [74] provide an algorithm that converges to an $\varepsilon$-approximate solution in $\tilde{O}(\mu_{\boldsymbol{x}}/\varepsilon)$ iterations, and Ouyang and Xu [60] provide a lower bound of $\tilde{\Omega}\left(\sqrt{\mu_{\boldsymbol{x}}/\varepsilon}\right)$ iterations on this same computation. Lin, Jin, and Jordan then went on to develop a faster algorithm, with iteration complexity of $\tilde{O}\left(\sqrt{\mu_{\boldsymbol{x}}/\varepsilon}\right)$, under the same conditions.

When $f$ is simply assumed to be convex-concave, Nemirovski [53], Nesterov [54], and Tseng [75] describe algorithms that solve for an $\varepsilon$-approximate solution with $\tilde{O}\left(\varepsilon^{-1}\right)$ iteration complexity, and Ouyang and Xu [60] prove a corresponding lower bound of $\Omega(\varepsilon^{-1})$.

When $f$ is assumed to be non-convex-$\mu_{\boldsymbol{y}}$-strongly-concave, and the goal is to compute a first-order Nash, Sanjabi et al. [67] provide an algorithm that converges to $\varepsilon$-an approximate solution in $O(\varepsilon^{-2})$ iterations. Jin, Netrapalli, and Jordan [41], Rafique et al. [64], Lin, Jin, and Jordan [45], and Lu, Tsaknakis, and Hong [47] provide algorithms that converge in $\tilde{O}\left(\mu_{\boldsymbol{y}}^2\varepsilon^{-2}\right)$ iterations, while Lin, Jin, and Jordan [46] provide an even faster algorithm, with an iteration complexity of $\tilde{O}\left(\sqrt{\mu_{\boldsymbol{y}}}\varepsilon^{-2}\right)$.

When $f$ is non-convex-non-concave and the goal to compute is an approximate first-order Nash equilibrium, Lu, Tsaknakis, and Hong [47] provide an algorithm with iteration complexity $\tilde{O}(\varepsilon^{-4})$, while Nouiehed et al. [58] provide an algorithm with iteration complexity $\tilde{O}(\varepsilon^{-3.5})$. More recently, Ostrovskii, Lowy, and Razaviyayn [59] and Lin, Jin, and Jordan [46] proposed an algorithm with iteration complexity $\tilde{O}\left(\varepsilon^{-2.5}\right)$.

When $f$ is non-convex-non-concave and the desired solution concept is a "local" Stackelberg equilibrium, Jin, Netrapalli, and Jordan [41], Rafique et al. [64], and Lin, Jin, and Jordan [45] provide algorithms with a $\tilde{O}\left(\varepsilon^{-6}\right)$ complexity. More recently, Thekumparampil et al. [74], Zhao [85], and Lin, Jin, and Jordan [46] have proposed algorithms that converge to an $\varepsilon$-approximate solution in $\tilde{O}\left(\varepsilon^{-3}\right)$ iterations.

We summarize the literature pertaining to the convex-concave and the non-convex-concave settings in Tables 3 and 4, respectively.

Table 3: Iteration complexities for min-max games (with independent strategy sets) in convex-concave settings. Note that these results assume that the objective function is Lipschitz-smooth.

| Setting | Reference | Iteration Complexity |
|---|---|---|
| $\mu_{\boldsymbol{x}}$-Strongly-Convex-$\mu_{\boldsymbol{y}}$-Strongly-Concave | [76] | $\tilde{O}\left(\mu_{\boldsymbol{x}} + \mu_{\boldsymbol{y}}\right)$ |
| | [83] | |
| | [30] | |
| | [51] | |
| | [3] | $\tilde{O}\left(\min\left\{\mu_{\boldsymbol{x}}\sqrt{\mu_{\boldsymbol{y}}}, \mu_{\boldsymbol{y}}\sqrt{\mu_{\boldsymbol{x}}}\right\}\right)$ |
| | [46] | $\tilde{O}(\sqrt{\mu_{\boldsymbol{x}}\mu_{\boldsymbol{y}}})$ |
| | [38] | $\tilde{\Omega}(\sqrt{\mu_{\boldsymbol{x}}\mu_{\boldsymbol{y}}})$ |
| | [84] | |
| $\mu_{\boldsymbol{x}}$-Strongly-Convex-Linear | [42] | $O\left(\sqrt{\mu_{\boldsymbol{x}}/\varepsilon}\right)$ |
| | [33] | |
| | [86] | |
| $\mu_{\boldsymbol{x}}$-Strongly-Convex-Concave | [74] | $\tilde{O}\left(\mu_{\boldsymbol{x}}/\sqrt{\varepsilon}\right)$ |
| | [46] | $\tilde{O}(\sqrt{\mu_{\boldsymbol{x}}/\varepsilon})$ |
| | [60] | $\tilde{\Omega}\left(\sqrt{\mu_{\boldsymbol{x}}/\varepsilon}\right)$ |
| Convex-Concave | [53] | $O\left(\varepsilon^{-1}\right)$ |
| | [54] | |
| | [75] | |
| | [46] | $\tilde{O}\left(\varepsilon^{-1}\right)$ |
| | [60] | $\Omega(\varepsilon^{-1})$ |

Table 4: Iteration complexities for min-max games (with independent strategy sets) in non-convex-concave settings. Note that although all these results assume that the objective function is Lipschitz-smooth, some authors make additional assumptions: e.g., [58] obtain their result for objective functions that satisfy the Lojasiwicz condition.

| Setting | Reference | Iteration Complexity |
|---|---|---|
| Nonconvex-$\mu_{\boldsymbol{y}}$-Strongly-Concave, First Order Nash Equilibrium or Local Stackelberg Equilibrium | [41] | $\tilde{O}(\mu_{\boldsymbol{y}}^2 \varepsilon^{-2})$ |
| | [64] | |
| | [45] | |
| | [47] | |
| | [46] | $\tilde{O}\left(\sqrt{\mu_{\boldsymbol{y}}} \varepsilon^{-2}\right)$ |
| Nonconvex-Concave, First Order Nash Equilibrium | [47] | $\tilde{O}\left(\varepsilon^{-4}\right)$ |
| | [58] | $\tilde{O}\left(\varepsilon^{-3.5}\right)$ |
| | [59] | $\tilde{O}\left(\varepsilon^{-2.5}\right)$ |
| | [46] | |
| Nonconvex-Concave, Local Stackelberg Equilibrium | [41] | $\tilde{O}(\varepsilon^{-6})$ |
| | [58] | |
| | [46] | |
| | [74] | $\tilde{O}(\varepsilon^{-3})$ |
| | [85] | |
| | [46] | |

# H   Future Directions

Our experiments suggest that the smoothness properties of the value function determine the convergence speed to a Stackelberg equilibrium. Additionally, our experiments with Leontief Fisher markets suggest that our convergence results could perhaps be generalized to convex-concave objective functions which are *not* necessarily continuously differentiable; such a result would require a generalization of the subdifferential envelope theorem we have introduced.

Another worthwhile direction would be to try to derive stronger conditions under which the value function $V$ is Lipschitz-smooth; under stronger assumptions we might be able to achieve faster convergence results. In order for $V$ to be differentiable, the subdifferential given by Theorem 3.2 would have to be a singleton, for all $\boldsymbol{x} \in X$. This would require not only that the solution function $Y^*(\boldsymbol{x})$ be a singleton for all $\boldsymbol{x} \in \mathcal{X}$, but that the optimal KKT multipliers $\Lambda(\widehat{\boldsymbol{x}}, \boldsymbol{y}^*(\widehat{\boldsymbol{x}}))$ were also unique. The former can be guaranteed when $f$ is strictly concave in $\boldsymbol{y}$, while the latter condition is satisfied when the Linear Independence Constraint Qualification condition [35] holds. However, even when both these conditions hold, and when both the objective function and the (parameterized) constraints (i.e., $\boldsymbol{g}(\boldsymbol{x}, \boldsymbol{y})$) are Lipschitz-smooth, the value function is not guaranteed to be Lipschitz-smooth.

Gao and Kroer [29] have proposed first-order methods for the efficient computation of competitive equilibria in large Fisher markets. It would be worth exploring whether our view of competitive equilibria in Fisher markets as min-max Stackelberg games can facilitate the extension of their fast convergence results to a larger class of Fisher markets.

A question of interest both for Fisher market dynamics and convex-concave min-max Stackelberg games is whether gradient-descent-ascent (GDA) converges in the dependent strategy set setting as it does in the independent strategy setting [45]. GDA dynamics for Fisher markets correspond to myopic best-response dynamics, which are of general economic interest (see, for example, [52]).

Finally, our results at present concern only *convex-concave* min-max Stackelberg games. It would be of interest to extend these results to the non-convex-concave setting. Doing so could improve our understanding of competitive equilibria in Fisher markets with non-homogeneous utility functions, and our understanding of optimal auctions. Recently, Dai and Zhang [16] defined the concept of a local minimax point, whose analog in Stackelberg games would be a local Stackelberg equilibria. We believe that any algorithm designed for a non-convex-concave setting would have to aim for a local solution concept, as it is unlikely that (global) Stackelberg equilibria are computable in polynomial time in non-convex-concave settings, as non-convex optimization is hard.

## Broader Impact

Our work can be used to expand the scope of many machine learning techniques as discussed in our introduction. For example, min-max optimization has been key to the development of fairer classifiers in recent years. On the other hand, our methods could also be used to train GANs, which have been used to create deepfakes with malicious goals. Our work can also be used to improve economic outcomes for companies running online marketplaces, as Fisher markets have been applied to resource allocation problems—specifically, fair division problems—in recent years. Moreover, with sufficient oversight, progress can be easily monitored, by following the trend of the the objective function across iterations, so that inappropriate use can be detected.