# OpenReview forum: "Convex-Concave Min-Max Stackelberg Games"
_NeurIPS.cc/2021/Conference — NeurIPS 2021 Poster_

### Official Review · Reviewer_4ud8 · 2021-07-10

**Rating:** 7
**Confidence:** 4

**Summary:**

This work introduces two first order (subgradient) methods (1 requiring an oracle) to solve convex-concave min-max problems with convex dependent feasible sets. The convergence of both algorithms is analyzed, and their performance is demonstrated on computing Stackleberg equilibria of a family of Fisher market models.


Contributions:
1) A subgradient algorithm assuming access to an oracle plus a nested subgradient algorithm (paired with explicit Fisher market counterparts) for computing Stackleberg equilibria in min-max problems with feasible sets
2) Convergence guarantees for the proposed algorithms
3) Adaptation of the general algorithms to the Fisher market domain
4) Empirical analysis of the proposed algorithms on problems from either side of the boundary designated by the algorithmic assumptions

**Limitations And Societal Impact:**

The authors mention the connection of their work to fairness and adequately address any possible negative impact.

**Main Review:**

Originality:
To my knowledge, these are the first subgradient algorithms for convex-concave min-max problems with convex dependent feasible sets.

Quality:
The paper is of high quality.

Clarity:
The paper is written very clearly. I only had one question around Theorem 3.3 below.

Significance:
The computation of equilibria in min-max games is generally important and connections to Fisher markets in economist are made. It’s possible additional connections to robust optimization could increase the paper’s significance.


Overall, I think the paper is well written and comprehensive. I think the paper could benefit from a few changes
1) A discussion of uniqueness of the solutions to the min-max problems.
2) Baselines and experiments
3) Potentially add robust optimization to related work


- [1] “Theory and Applications of Robust Optimization” Bertsimas et al. 2011
- [2] “Online first-order framework for robust convex optimization” Ho-Nguyen and Kılınç-Karzan. 2016


1) Uniqueness:
One part of Theorem 3.3 appears suspect. It is claimed that Algorithm 1 converges asymptotically to $(x^*, y^*)$ which is a $(0, \delta)$-Stackleberg equilibrium, but is $(x^*, y^*)$ a unique equilibrium? I would assume there are many such $(0, \delta)$-equilibria. Can you please address this and add a discussion around uniqueness of solutions for these minmax problems with dependent feasible sets? Maybe you meant to say that Algorithm 1 converges to one of the many $(0, \delta)$-equilibria, not a specific one.


2) Baselines and Experiments:
- After reading the related work section, I was expecting an empirical comparison to the interior point methods of Ye [33] and first-order methods of Gao & Kroer [23]. Is there a reason you did not compare to them? The paper would benefit from having baselines to compare against.
- Didn’t you need to set a value for $\epsilon$ or $\delta$ for the inner loop? Ah, I see you opted for number of iterations $T_X$ instead. I’d be interested to know the ratio of $T_X / T_p$ used in the experiment. Also, I would like to see a sweep over this parameter, which could provide further insight into understanding at which point GDA might fail (low values of the ratio move Alg 4 closer to alternating ascent descent which is closer to GDA).


3) Robust Optimization:
At first glance, the min-max problem with dependent feasible sets seems related to robust optimization. Briefly, see https://en.wikipedia.org/wiki/Robust_optimization, https://en.wikipedia.org/wiki/Wald%27s_maximin_model or [1] and [2] for first-order online approaches.


Minor:
- Are the $g_i$ required to be concave so that the intersection with $Y$ forms a convex set? Is this the most general setting?
- A figure illustrating how taking a step changes the feasible set and hence presents problems for GDA could be enlightening and convince the reader of the difficulty of the problem
- line 60: After a double-take, I understand what you’re saying but saying $V(x)$ is $y$’s problem is confusing as $y$ doesn't appear as an argument. It’s probably better to write out "player $y$'s problem is $\max_{y \in Y}:g(x,y) \ge 0 f(x,y) = V(x)$" again.
- line 98: does $\epsilon$-approximation refer to error in the strategy space or value function space or both? It appears value function space (Def 2.1). It would help the reader if you mentioned this around line 98.
- Table 2, strongly concave case — I didn’t check the proof, but shouldn’t $\log(\epsilon)$ be $\log(\frac{1}{\epsilon})$? Seems like a typo.
- Figure 1 - maybe refer to the algorithms with a description as well since they are not included in the main body, e.g., Algorithm 3 (oracle, in blue) and Algorithm 4 (nested, in orange). Also, why not plot on a log scale?

**Time Spent Reviewing:**

2

---

> ### Author Response · Authors · 2021-08-10
> **Answer to reviewer 4ud8**
>
>
> Firstly, many thanks for your thorough and insightful review!
>
> Regarding your comments and proposed improvements:
>
> ***Uniqueness***:
>
> The value of $(0,0)$-Stackelberg equilibria, i.e., the equilibrium payoffs of the players, are unique, however the optimal strategies supporting a $(0,0)$-Stackelberg equilibrium are not unique. And as you noted there are many $(0,\delta)$-Stackelberg equilibria. We will make the necessary changes in our paper to address your comment.
>
>
> ***Baselines and Experiments***:
>
> [1] “First-Order Methods for Large-Scale Market Equilibrium Computation”, Yuan Gao and Christian Kroer, 2020
>
> - As Gao and Kroer mention [1], interior point methods (IPMs) outperform first-order methods (FOMs) when computing Fisher market equilibria; however, they become unstable in large markets, as they involve solving large systems of linear equations. Thus, following Gao and Kroer, we also opted to not use IPMs as baselines. Additionally, the min-max program with dependent feasible sets for Fisher markets couples the computation of equilibrium allocations and prices. This means that once our algorithm converges, it has found a complete market equilibrium, while the algorithms proposed by Gao and Kroer compute either equilibrium prices or equilibrium allocations but not both, which means that additional non-trivial computation would be required.
>
> - Your point is very interesting and we believe that your suggestion would be very valuable for future work in understanding the convergence properties of GDA.
>
> Space permitting, we would be happy to include additional experiments! Thank you for the suggestions.
>
> ***Robust Optimization***:
>
> [2] “Online first-order framework for robust convex optimization”, Ho-Nguyen and Kılınç-Karzan. 2016
>
> [3] “Oracle-Based Robust Optimization via Online Learning”, Ben-Tal et al. 2014
>
> [4] ”First-order algorithms for robust optimization problems via convex-concave saddle-point Lagrangian reformulation” Krzysztof Postek, Shimrit Shtern, 2021
>
> Thanks for this suggestion. We agree that robust optimization (RO) should appear in the related work section. The problem of min-max games with dependent feasible sets can be seen as equivalent to RO (although the min-max game with dependent strategy sets framework is slightly more general in its statement). In the cursory review of the RO literature that we managed to do since receiving your review, we observed that recent approaches developed to solve RO require a transformation of the problem into a feasibility problem ([2], [3]), and as pointed out by [4], these methods either require oracles and are stochastic in nature [3], or require binary search for the optimal value, which can be computationally complex [4]. We believe this might be because these methods follow the gradient of the value function given by Danskin’s theorem, which is the incorrect gradient to follow, and thus require that they use information other than just the gradient to find an optimal solution. Our methods can potentially be used to solve a large class of convex RO problems, circumventing these issues, as they take a Lagrangian-based saddle-point approach similar to [4], and so, do not require us to convert the problem into a feasibility problem. This means that our algorithms can be used for RO to extend the convergence guarantees of [4] from linear RO problems with convex uncertainty sets to a large class of convex RO problems with convex uncertainty sets. We also note that these methods [2] return only $\epsilon$-feasible points, while our method guarantees that any output will always be feasible, which is more desirable in practice. Our theoretical results also provide more insights into the properties of the optimization problem that affect the convergence rate, as compared to previous work [2, 3, 4]. For instance, [4] provides convergence rates only when the objective function is Lipschitz, while we provide faster convergence guarantees when the objective function is additionally strongly-convex-strongly-concave.
>
> ***Regarding your minor comments***:
>
> - The $g_i$’s are required to be concave so that the intersection with Y forms a convex set. This in turn ensures that the KKT theorem holds, so that the subdifferential envelope theorem can be derived. This is the most general setting we can envision for the moment, as non-concave $g_i$’s would require the development of additional theoretical tools.
> - We will be thinking of such a figure and aim to add it to the paper. Thank you for the suggestion!
> - We will make the necessary modifications.
> - The $\epsilon$-approximation is in value space; we will add the necessary clarifications.
> - Yes, that is a typo. It should indeed be $\log(\frac{1}{\epsilon})$! Thank you for noticing this. We will make the necessary modifications.

---

### Official Review · Reviewer_Krbq · 2021-07-16

**Rating:** 7
**Confidence:** 3

**Summary:**

The paper studies min-max optimization problems with dependent strategy sets, and introduces a first-order method that solves a large class of convex-concave min-max games with dependent strategy sets. In doing so, the authors provide a suitable envelope theorem which generalizes known results to the case of dependent strategies. That result is used to build two algorithms: max-oracle gradient-descent (which returns an approximation of the inner player's value function), and nested gradient-descent/ascent (which computes both players' strategies explicitly). The algorithms are shown to converge in polynomial time, and  they are experimentally evaluated through the computation of competitive equilibria in Fisher markets with varying utility structures.

**Ethical Concerns:**

No ethical concerns

**Limitations And Societal Impact:**

See main review

**Main Review:**

The paper is well written and mostly clear. The general problem of minmax games with dependent strategy sets is well motivated by applications such as computing equilibria in Fisher markets. The set of techniques presented may be a good starting point for future works on other related problems.

I'd be interested in hearing more about how the method proposed by the authors compares (in the Fisher markets setting) to other recent works such as Gao and Kroer [23].



**Time Spent Reviewing:**

3

---

> ### Author Response · Authors · 2021-08-10
> **Answer to reviewer Krbq**
>
> Thank you for your review!
>
> First, we would like to note that in theory market equilibria in Fisher markets with continuous, concave, and homogeneous utility functions could be computed fast by solving the Eisenberg-Gale program via interior point methods that handle exponential cones. However, these methods do not scale to large markets since interior point methods require solving a very large linear system of equations. This makes first-order methods such as ours attractive for the computation of market equilibria.
>
> We now reiterate here some of what we replied to reviewer CPGS.
>
> There are two advantages to using the min-max framework to compute Fisher market equilibria, as compared to the methods used by Gao and Kroer. First, the min-max framework couples the computation of equilibrium allocations and prices. As a result, once our algorithm converges, it has found a complete market equilibrium, while the algorithms proposed by Gao and Kroer compute either prices or allocations but not both, which means that additional non-trivial computation is required. For instance, in the case of linear utilities Gao and Kroer show that their first-order methods converge to equilibrium prices, but to obtain an equilibrium allocation one would need to solve an additional optimization problem for each buyer.
>
> Second, their theoretical results only apply to Fisher markets with linear, quasilinear, and Leontief utilities, while our min-max formulation allows us to derive convergence bounds in Fisher markets with any continuous, concave, and twice differentiable utility functions, a class of utility functions infinitely larger than that considered by Gao and Kroer. Prior to our work, there were no theoretical guarantees of convergence of first-order methods in such a large class of markets, i.e, Fisher market with twice differentiable utility functions; all results were focused on specific classes of utility functions, e.g., linear utilities.

---

### Official Review · Reviewer_gWmB · 2021-07-17

**Rating:** 6
**Confidence:** 3

**Summary:**

The authors study sequential (aka Stackelberg) games, where the actions of the follower are subjects to constraints depending on the leader's action.
They derive a formula for the subgradient of leader's utility (under optimal action of the follower) and use it to derive subgradient descent algorithms with convergence guarantees.

**Limitations And Societal Impact:**

yes

**Main Review:**

The authors study minimization problems of the form $\min \limits_{x} V(x) = \min \limits_{x} \max \limits_{y: g(x, y) \geq 0} f(x,y)$
for f convex-concave and g component-wise concave.
The convex function $V(x)$ is convex but in general only subdifferentiable.
If $V$ is differentiable in $x$, recent results by Milgrom and Segal allow to express its gradient in terms of the optimal solution $y, \lambda$ of the inner loop, where $\lambda
$ is a lagrange multiplier associated to the constraint.
The authors extend this results to a characterization of the subgradient of $V(x)$ in arbitrary points, and derive rates for the resulting first order methods, accounting also fo
r the solution of the inner loop.
They furthermore test their algorithms on a series of Fisher markets.

This appears to be solid work, although I am not sure about the relevance of the author's results to the rest of the NeurIPS community.
The authors do not provide any explanation to this effect, with the exception of line 376, which is fairly vague (in which GAN model do dependent strategy sets arise?).
Venues more focused on economics or convex optimization might also be better positioned to appreciate the theoretical contribution of the Subdifferential Envelope Theorem.

I am therefore somewhat on the fence about this paper, maybe slightly leaning towards rejection but open to being convinced otherwise in the rebuttal.

Comments to authors: As illustrated in the paper, convex games with dependent strategy sets do not satisfy a minimax theorem in general.
However, they can also be interpreted as simultaneous (non-\{sequential/stackelberg\}) non-zero-sum games (with rewards attaining $-\infty$).
I might be helpful to change the title and possibly introduction to clarify that the paper deals with sequential games.

**Time Spent Reviewing:**

4

---

> ### Author Response · Authors · 2021-08-10
> **Answer to reviewer gWmB**
>
> [1] “Optimal Auctions Through Deep Learning”, Dütting et al., 2021
>
> [2] ”First-order algorithms for robust optimization problems via convex-concave saddle-point Lagrangian reformulation” Krzysztof Postek, Shimrit Shtern, 2021
>
> [3] Chen, R. S., Lucier, B., Singer, Y., and Syrgkanis, V. Robust optimization for non-convex objectives. In Advances in Neural Information Processing Systems, pp. 4705–4714,
> 2017.
>
>
> Thank you for your review! We would like to address your concern that our paper might better be directed to an economics audience. We submitted this project to NeurIPS because most work on first-order methods for min-max optimization in recent years has appeared at NeurIPS, and in particular, so did the very related work by Gao and Kroer on first-order methods to solve Fisher markets. Additionally, we note that in most cases of theoretical interest in mathematical economics, researchers restrict themselves to value functions which are differentiable or ignore the non-differentiability issue entirely, since the value function is absolutely continuous and as a result differentiable almost everywhere. Consequently, the subdifferential envelope theorem might be of greater interest to computer scientists than to economists, since even if the value function is differentiable almost everywhere, algorithms might land on non-differentiable points. Furthermore, the subdifferential’s properties might provide additional insights about the value function. For instance, one may be able to prove that the value function is strongly convex using information about the subdifferential of the value function. This in return can allow for better convergence guarantees of first-order methods when optimizing such value functions. As a result, we believe that this result is more relevant to a computer science audience than an economics audience, especially when taking in consideration the two following applications:
>
> First, we want to bring to your attention an application of our min-max games with dependent feasible sets to deep learning which was brought to our attention after this paper was submitted, specifically in the 2021 edition of Communications of the ACM, where Dütting et al. [1] propose a neural network architecture called RegretNet to learn optimal auctions. The learning objective of this neural network corresponds to a min-max game with dependent strategy sets. That is, the RegretNet learning objective can be seen as a min-max Stackelberg game between a fictional auctioneer (note that in this case the auctioneer picks the weights of the neural network which determines the pricing and allocation rules) and the bidders. In particular, the auctioneer aims to minimize its expected negated revenue (i.e., maximize expected revenue) while the bidders aim to maximize the auctioneer’s expected negated revenue (i.e., minimize expected revenue) constrained by a no-regret condition. More interestingly, the algorithm they use to train their neural network corresponds exactly to the nested gradient descent ascent algorithm we propose. Whereas they provide no convergence guarantees in their paper, we do provide guarantees under certain assumptions (which their network does not satisfy). We believe that in the coming years more applications of our algorithm and close variants will emerge for Deep Learning problems. Initiating the study of algorithms to train architectures with such learning objectives will prove crucial to improving our understanding of these neural networks.
>
> Finally, we want to bring to your attention an interesting application of our algorithms to robust optimization (RO) problems as pointed out by reviewer 4ud8. RO is an optimization framework which ensures that the solution of a problem is optimal for all possible realizations of the problem given by parameters in an uncertainty set. RO can be seen as a min-max game with dependent feasible sets, in which the first player is the optimizer, and the second player is nature. The optimizer picks an optimal solution among all possible realizations of nature in the uncertainty set, which in return affects the uncertainty set, and then nature picks a realization of the world from the ensuing uncertainty set. Our algorithms can be used to solve a large class of convex RO problems with a convex uncertainty set, improving upon Postek and Shtern’s 2021 results [2], which apply only to linear RO problems with convex uncertainty sets. We will include relevant discussion and references to make the connection between robust optimization and min-max games with dependent feasible sets, which should make this work even more relevant to the NeurIPS audience (e.g.,  [3]).

---

> > ### Comment · Reviewer_gWmB · 2021-09-01
> > **Thank you for the additional references**
> >
> > I am still on the fence about the paper, but am willing to raise my score to 6. I recommend that the authors try to provide a bit more machine learning motivation in the revised version.

---

> > > ### Author Response · Authors · 2021-09-02
> > > **Thank you!**
> > >
> > > Thank you for your feedback and score increase, we will make sure to address your concerns!

---

### Official Review · Reviewer_CPGS · 2021-07-19

**Rating:** 7
**Confidence:** 3

**Summary:**

This paper studies the minmax problem where the second player's (follower) action set depends on the action of the first player. Naturally, Stackelberg equilibrium is the concept to study here. The authors use a sub-gradient formulation of the known envelope theorem to solve the problem and experiment on Fisher market set-up.


**Limitations And Societal Impact:**

Yes.

**Main Review:**

In my reading of the paper, the work felt incremental because of
1) The algorithm as such is exactly (for Alg 1) or mostly (for Alg 2) in the style of Jin et al. (or actually, papers before that also) with the difference being that the sub-gradient is used. Hence, the convergence proof also mostly follows standard sub-gradient convergence proofs.
2) Fisher market and the Eisenberg Gale convex form has already been shown to admit first order methods on the direct EG form in Gao and Kroer. So, not sure why the same problem needs to be solved in a different way and is there any advantage of this first order form over that? The authors leave this future work but this comparison should be done for this work.
3) More details on experiments (how the utility defining parameters were chosen instead of just higher and lower values as in Fig.1 ) needs to be in the main paper

Writing:
Easy to read, short simple examples given to assist the readers in understanding the problem with and without dependent sets.
The introduction is just too long, suggest breaking it up into more sections.

**Time Spent Reviewing:**

2.5 hours

---

> ### Author Response · Authors · 2021-08-10
> **Response to Reviewer CPGS**
>
> Thank you for your review! As shown in Example 1.2, algorithms of the type in Jin et al. do not converge to the solutions of min-max games with dependent strategy sets. Further, the main difference between their approach and ours is not simply that we use the subgradient instead of the gradient, but rather that we use the value function's (sub)gradient, instead of the objective function's gradient, the latter of which we show by counterexample is not the correct gradient for our problem. The reason we use the subgradient instead of the gradient is simply that the value function of the outer player is not guaranteed to be differentiable. The fact that we were able to provide theoretical convergence guarantees in such a large setting with such a simple algorithm for a problem that was first considered in 1981 by Shimizu and E. Aiyosh is, in our view, a strength rather than a weakness. In sum, no previous first-order method can solve our problem and algorithms that have been proposed in the last 40 years provided no convergence guarantees, while we were able to provide guarantees with a simple iterative first-order method. Additionally, even with access to our novel subdifferential envelope theorem, convergence of our method does not follow from results on the convergence of subgradient methods when we can only achieve an epsilon optimal solution to the inner player’s objective function (which is the best we can hope for in practice). That is the purpose of our Theorem 3.3, as we are not guaranteed that the epsilon terms will not accumulate over time, leading to a blow-up in the payoffs and eventual non-convergence. (Note that Lemma D.1, which asserts that the value function is Lipschitz continuous when the objective function is Lipschitz, provides some intuition for this theorem, but in and of itself it is not sufficient. We include it only for intuition; it plays no role in the convergence proofs.) Finally, our paper provides the first proof of equivalence between a large class of competitive equilibria and Stackelberg equilibria, the first result of its kind that we are aware of, as previous results have proven equivalence between Nash equilibria and competitive equilibria in some markets. This novel equivalence result provides important intuition on the computational complexity of market equilibrium problems.
>
> There are two advantages to using the min-max framework to compute Fisher market equilibria, as compared to the methods used by Gao and Kroer. First, the min-max framework couples the computation of equilibrium allocations and prices. As a result, once our algorithm converges, it has found a complete market equilibrium, while the algorithms proposed by Gao and Kroer compute either prices or allocations but not both, which means that additional non-trivial computation is required. For instance, in the case of linear utilities, Gao and Kroer show that their first-order methods converge to equilibrium prices, but to obtain an equilibrium allocation one would need to solve an additional optimization problem for each buyer. Second, their theoretical results only apply to Fisher markets with linear, quasilinear, and Leontief utilities, while our min-max formulation allows us to derive convergence bounds in Fisher markets with any continuous, concave, and twice differentiable utility functions, a class of utility functions infinitely larger than that considered by Gao and Kroer. Prior to our work, there were no theoretical guarantees of convergence of first-order methods in such a large class of markets, i.e, Fisher market with twice differentiable utility functions; all results were focused on specific classes of utility functions, e.g., linear utilities.
>
> We will be sure to move more detailed information about the experiments into the main body of the paper, as you and other reviewers have suggested.

---

### Decision · Program_Chairs · 2021-09-27

**Decision:**

Accept (Poster)

**Comment:**

This paper studies a class of min-max games where each player's actions are subject to coupled constraints, i.e., the actions avaible to one player depend on the action chosen by the other, and vice versa. The authors examine two sequential algorithms for solving the game - one based on best responses and another based on a nested version of gradient descent - and they derive a series of polynomial complexity guarantees for the algorithms under study.

The reviewers appreciated the contributions of the paper and were initially positive. However, during the discussion phase, it became clear that the paper is ignoring an extensive literature on generalized Nash equilibrium problems. This literature dates back at least to Debreu's seminal 1952 paper and it treats essentially the same problem as the authors: in particular, there is a series of asymptotic convergence results (for different classes of games) that should be discussed by the authors. More precisely, even though the literature on learning in generalized Nash equilibrium problems does not provide rates of convergence, the results provided therein prove the convergence of the actual sequence of play (something which does not seem possible with the authors' approach and techniques).

Another - more minor - concern had to do with the relevance of such models for the ML/AI community, but this was partially addressed by the authors.

Overall, the paper definitely has merits: to the best of the committee's knowledge, this is the first polynomial complexity result of its kind in the class of games with coupled constraints (which is perhaps not surprising since the literature on generalized Nash equilibrium problems otherwise focuses on the convergence of the actual sequence of play, not the "best" or "time-averaged" iterate). However, the lack of proper positioning is a major issue that has to be fixed for the camera-ready version; specifically, the committee has the following expectations:
1. The authors should provide adequate pointers to the literature on generalized Nash equilibrium problems, especially on early work by Facchinei and co-authors, as well as more recent contributions by Grammatico and co-authors - e.g., the recent preprint by Fabiani et al., "Local Stackelberg equilibrium seeking in generalized aggregative games" should be discussed in detail. [Certain references were already provided during the discussion phase, and the authors would be expected to expand on those]
2. The authors should present in detail the notion of generalized Nash equilibria in the paper (during the discussion phase, there seemed to be some confusion on this point by the authors, but it was eventually cleared up). In particular, the committe expects the authors to provide an in-depth comparison to the model of Fabiani et al. - not in regard to the _class of games_ considered, but as to the _notion of equilibrium_ under study. Put simply, now that the authors are aware of the literature on the problem, the committee would expect an in-depth and balanced treatment.
3. The algorithms studied by the authors should be incorporated in the main text - otherwise, the statements of the authors' theorems are impossible to parse.

Modulo the above, I am happy to recommend acceptance.